# Structural insights into the activation of the chicken ROS1 receptor by the NEL/NICOL ligand complex

Weidong An [1], Xuewu Zhang [1,2] ✉ & Xiao-chen Bai [1,3] ✉

The receptor tyrosine kinase ROS1 plays essential roles in cell growth and sperm maturation, yet its activation mechanism has remained poorly understood. Here, we report high-resolution cryo-electron microscopy (cryo-EM) structures of chicken ROS1 in its ligand-free form, in complex with its ligand NEL, and with the ligand/co-ligand complex NEL/NICOL. Unliganded ROS1 adopts an arc-shaped conformation. The interaction between NEL and ROS1 is mediated by the VWC2 domain of NEL and the β1 domain of ROS1. Binding of NICOL to the coiled-coil domain of NEL stabilizes NEL into a batwing-shaped asymmetric dimer, which can recruit only one ROS1 molecule due to steric hindrance. Structural analyses and biochemical results suggest that the 2:1 NEL/NICOL complexes further oligomerize through LamG–VWC4 domain interactions, facilitating the clustering of multiple ROS1 for its activation. Functional assays confirm that both NICOL and the multimerization of NEL/NICOL complexes are required for robust ROS1 signaling. Our findings establish NICOL as a critical co-ligand for ROS1 and suggest a distinct ligand-driven oligomerization mechanism for ROS1 activation.

The *ROS1* gene is evolutionarily conserved across a broad range of animal species, from early-diverging metazoans to mammals[1]. It encodes a receptor tyrosine kinase (RTK)[2,3], a type of transmembrane receptor that plays vital roles in regulating animal growth and development. In higher vertebrate species, ROS1 is expressed in various tissues, including testis, lung, heart, intestine, and kidney[4–7], and has been implicated in various physiological processes, including regulating growth and development[1]. Recent research has indicated that ROS1 signaling is required for sperm maturation and male fertility[8].

Aberrant expression, mutations, and somatic chromosomal fusions of the *ROS1* gene have been implicated in a variety of diseases, including both cancerous and non-cancerous conditions[1,9]. Dysregulated ROS1 expression—such as overexpression, splice variants, and gene amplification—can drive oncogenesis across multiple tissues and organs, including the lung, breast, stomach, liver, kidney, and colon[1,9]. Notably, ROS1 frequently fuses with other oncogenes to form ROS1 fusion proteins, which act as potent oncogenic drivers[10–14]. Over 50 different 5′ gene partners have been identified that fuse with the 3′ region of ROS1[9]. These fusion partners typically harbor dimerization domains, leading to constitutive, ligand-independent activation of ROS1's kinase activity[1,9,14]. In addition to its oncogenic roles, mutations in ROS1 have also been associated with cardiovascular disorders[1].

ROS1 is the largest RTK identified to date[2]. Structurally, like other RTKs, ROS1 consists of an extracellular domain (ECD), a transmembrane (TM) helix, and an intracellular kinase domain (KD). The ROS1 ECD is predicted to include a cysteine-rich ALKAL-type coupled helices domain (CATCH), nine fibronectin type III domains (FnIII-1–FnIII-9), and three β-propeller domains (β1–β3) (Fig. 1a). Recent cryo-electron microscopy (cryo-EM) studies have reported structures of Sevenless (Sev), the *Drosophila* ortholog of vertebrate ROS1[15,16]. These studies determined the structures of the ligand-free-Sev, the ligand (BOSS)-bound Sev, and dimeric Sev at low pH. Although these structures shed

[1]Department of Biophysics, University of Texas Southwestern Medical Center, Dallas, TX, USA. [2]Department of Pharmacology, University of Texas Southwestern Medical Center, Dallas, TX, USA. [3]Department of Cell Biology, University of Texas Southwestern Medical Center, Dallas, TX, USA. ✉e-mail: xuewu.zhang@utsouthwestern.edu; xiaochen.bai@utsouthwestern.edu

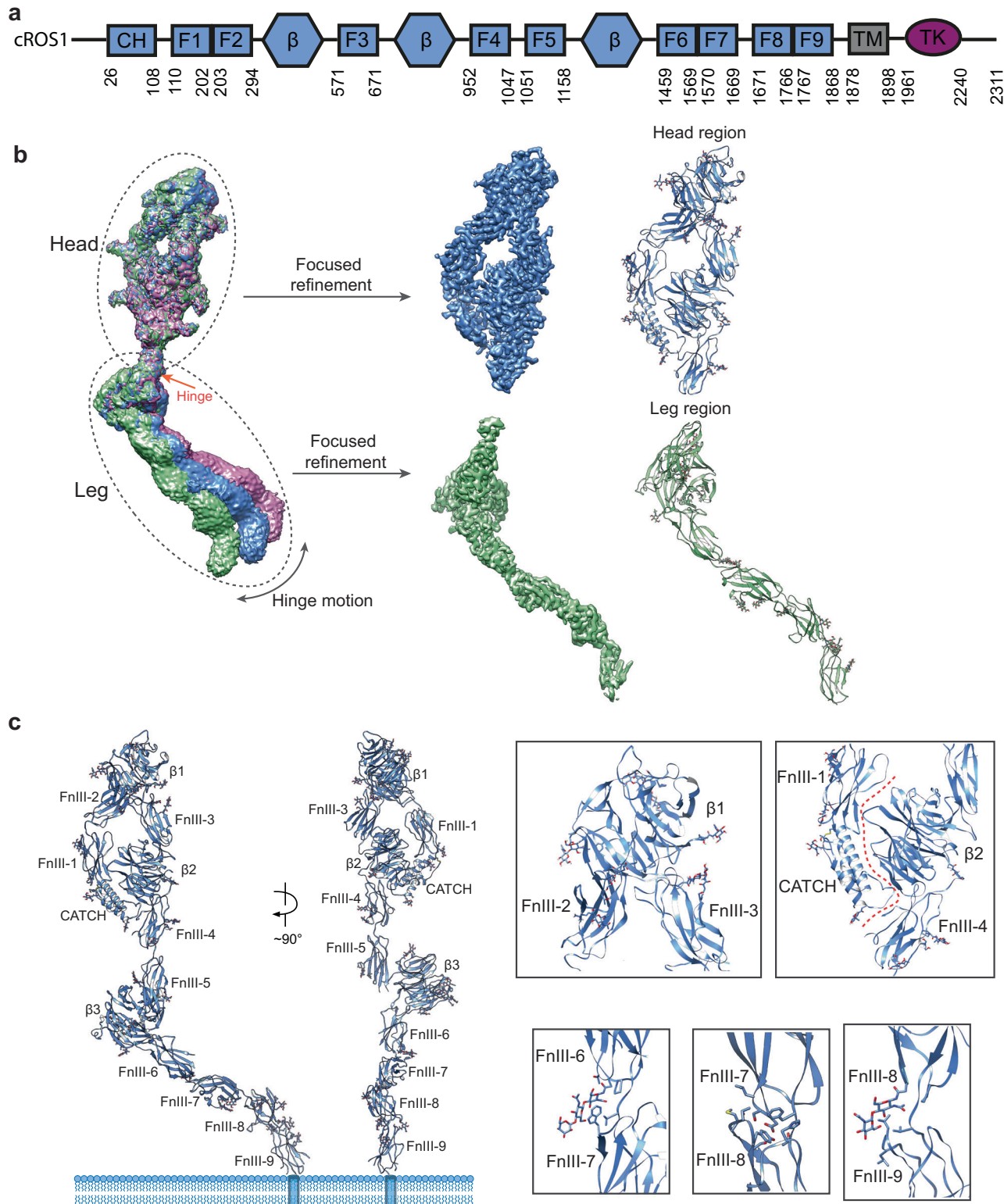

**Fig. 1 | Overall structures of ligand-free-cROS1. a** Domain structures of chicken ROS1. **b** The cryo-EM structure of ligand-free-cROS1 shows the conformational flexibility of the "leg" region relative to the "head" region. The 3D reconstructions of the "head" and "leg" parts of the ligand-free-cROS1 after focused 3D refinement at 2.9 Å and 3.6 Å resolution, respectively, and the corresponding ribbon representation of these two parts, are shown. **c** The ribbon representations of complete ligand-free-cROS1 are shown in two orthogonal views. The detailed domain-domain interactions in the ligand-free-cROS1 are shown in the close-up views. N-glycans play important roles in mediating the domain-domain interactions of ligand-free-cROS1.

light on the three-dimensional domain organization of the ROS1 ECD, they offer limited insight into ligand recognition and signal transduction mechanisms of ROS1 in higher vertebrates.

Vertebrate ROS1 had long been considered an orphan receptor, until a study in 2020 reported that neural epidermal growth factor–like 2 (NELL2) specifically binds to the extracellular domain of mouse ROS1[8]. This interaction was shown to activate the ERK signaling pathway, which is essential for the differentiation of the initial segment of the caput epididymis and ultimately critical for sperm maturation. Follow-up work by the same group identified NICOL (NELL2-interacting cofactor for lumicrine signaling), a small secreted protein, is necessary for ROS1/NELL2–mediated ERK activation[17,18]. However, the precise mechanism by which NELL2 and NICOL together activate ROS1 remains unclear.

Here, we determine the cryo-EM structures of the ligand-free form of chicken ROS1 (cROS1), the cROS1–chicken NEL (cNEL, homologous to human NELL2) complex, and the cROS1/cNEL/hNICOL complex. In its ligand-free state, cROS1 adopts an arc-shaped conformation. The binding of cNEL is mediated by the VWC2 domain of cNEL and the β1 domain of cROS1. The cNEL/hNICOL complex forms an asymmetric, batwing-like structure, in which the coiled-coil (CC) domains of the two cNEL protomers and α-helical region of hNICOL together form a central three-helix bundle. Despite the presence of two cNEL protomers in this complex, it can only engage one cROS1 molecule at a time due to steric hindrance. Further structural and functional analyses suggest that ROS1 activation occurs through higher-order oligomerization of the ROS1/NEL/NICOL complex, bringing multiple ROS1 receptors into proximity and enabling trans-autophosphorylation.

## Results

### Structure determination of chicken ROS1 in the ligand-free-state

To investigate the structure of vertebrate ROS1 in its unliganded state, we purified the full ECD of chicken ROS1 (cROS1), which exhibits high expression levels and shares 54% sequence identity with human ROS1 (Supplementary Fig. 1). Initial cryo-EM analysis of the cROS1-ECD revealed an arc-shaped architecture comprising two rigid parts, referred to as the head and leg regions respectively (Supplementary Fig. 2). Subsequent 3D classification identified a subtle hinge-motion between these two regions, with the angle between them ranging from 120° to 150° (Fig. 1b). Focused refinement enabled us to resolve the head and leg regions at resolutions of 2.9 Å and 3.6 Å, respectively, allowing accurate modeling of nearly the entire cROS1 ECD, spanning residues 24 to 1865 (Supplementary Fig. 2).

### Overall architecture of ligand-free cROS1

The head region of cROS1 consists of the CATCH, the first four FnIII domains (FnIII-1 to FnIII-4), and the first two β-propeller domains (β1 and β2) (Fig. 1b, c). The CATCH is a domain not typically found in other RTKs and is composed of four α-helices stabilized by multiple disulfide bonds. Within the head, the β1–FnIII-3–β2–FnIII-4 segment folds back and aligns anti-parallelly with the CATCH–FnIII-1–FnIII-2 segment, forming a compact, "D"-shaped architecture (Fig. 1b, c). This conformation is stabilized by extensive electrostatic and hydrophobic interactions—specifically between FnIII-2 and β1–FnIII-3, as well as between the CATCH–FnIII-1 and β2 (Fig. 1c). Additionally, a loop from the CATCH domain inserts into a cavity formed by β2 and FnIII-4, further reinforcing the rigidity of the head structure (Fig. 1c). A similar D-shaped fold has been observed in the structure of Sevenless (Sev), the *Drosophila* ortholog of ROS1, indicating the structural conservation of ROS1 across species[15,16].

In contrast to the compact head region, the leg region of cROS1—comprising the FnIII-5 to FnIII-9 domains and the β3 domain—adopts an extended, linear conformation spanning approximately 200 Å (Fig. 1b). Within this region, the FnIII-5 domain packs closely against β3 but forms only a weak association with the adjacent FnIII-4 domain of

the "head," accounting for the observed flexibility between the head and leg regions of cROS1. Notably, several N-glycans are positioned at the interfaces between consecutive domains from FnIII-6 to FnIII-9, enhancing interdomain interactions and contributing to the high rigidity of the "leg" (Fig. 1c).

### Structure determination and overall architecture of cROS1 in the cNEL bound state

To investigate the activation mechanism of ROS1 by NEL, we purified the chicken NEL protein (cNEL), which shares 88% sequence identity with human NELL2, using HEK293F cells. Purified cNEL primarily exists as a disulfide-linked dimer, with a small fraction present as a trimer[19,20] (Supplementary Fig. 1). Interestingly, a previous study indicates that recombinant human NELL1 and NELL2 purified from insect cells predominantly exist as trimers[21]. The observed differences in oligomeric state may have arisen from differences in the expression system or from species-specific variation. We first analyzed the binding affinity between cNEL and cROS using biolayer interferometry (BLI). The results showed that cNEL binds cROS1 strongly, with the $K_d$ value in the single-digit nanomolar range (~2 nM) (Supplementary Fig. 3). Subsequently, we reconstituted the cROS1/cNEL complex for cryo-EM structural analysis (Fig. 2a and Supplementary Figs. 1, 4, 5a). Co-elution of cROS1 and cNEL during size-exclusion chromatography (SEC) indicated the formation of a stable complex (Supplementary Fig. 1). Initial 3D reconstruction of the cROS1/cNEL complex was resolved at 2.6 Å resolution, revealing the complete head region of cROS1 and the VWC2 domain of cNEL. Subsequent 3D classification allowed the visualization of the VWC1 domain and the first two EGF domains (EGF1–2) of cNEL, though at relatively low resolution (Fig. 2b, c and Supplementary Fig. 4).

Our cryo-EM model reveals that the VWC1–VWC2–EGF1–EGF2 domains adopt a pin-like architecture (Fig. 2c). VWC1 interacts directly with EGF1, stabilizing the pin-like fold, while VWC2 is positioned at the tip of the pin, facilitating its interaction with cROS1. The remainder of the cNEL structure was unresolved, likely due to substantial conformational flexibility. Comparative structural analysis between the cROS1/cNEL complex and ligand-free-cROS1 indicates that cROS1 does not undergo significant conformational changes upon complex formation with cNEL. Given that cNEL predominantly forms a flexible disulfide-linked dimer, a single cNEL dimer may be capable of binding two cROS1 molecules simultaneously, although such dimeric complex is not resolved in our cryo-EM analysis.

### The interactions between cROS1 and cNEL

In the cROS1/cNEL complex, cNEL is positioned atop the head region of cROS1, with the interaction interface formed between a lateral surface of the β1 domain of cROS1 and a major β-sheet within the VWC2 domain of cNEL (Fig. 2b–d). Specifically, hydrophobic interactions between Phe336, Val349, Val357, and Phe359 of cNEL and Phe318, Leu319, and Val543 of cROS1 are critical for mediating the cROS1/cNEL association (Fig. 2d). Additionally, a salt bridge forms between Arg368 of cNEL and Glu541 and Asp545 of cROS1. Several other residues—Ser351, Lys365, and Gln367 of cNEL, along with Arg524, Phe527, Glu530, Ser538, Asn540, Pro551, and Tyr553 of cROS1—also contribute to the stability of the interface (Fig. 2d).

To validate the functional importance of this interface, we introduced F318A, E541A, F527A, and Y553A point mutations into the β1 domain of cROS1. These cROS1 mutants exhibited reduced activity in response to cNEL stimulation (Fig. 2e). Together, these findings underscore the critical role of the VWC2–β1 interface in cROS1 activation.

### The cryo-EM structure of cROS1/cNEL/hNICOL1 complex

NICOL has been shown to enhance NEL-dependent ROS1 activation by directly associating with NEL[17]. To investigate the underlying mechanism, we co-expressed cNEL with either chicken NICOL(cNICOL)

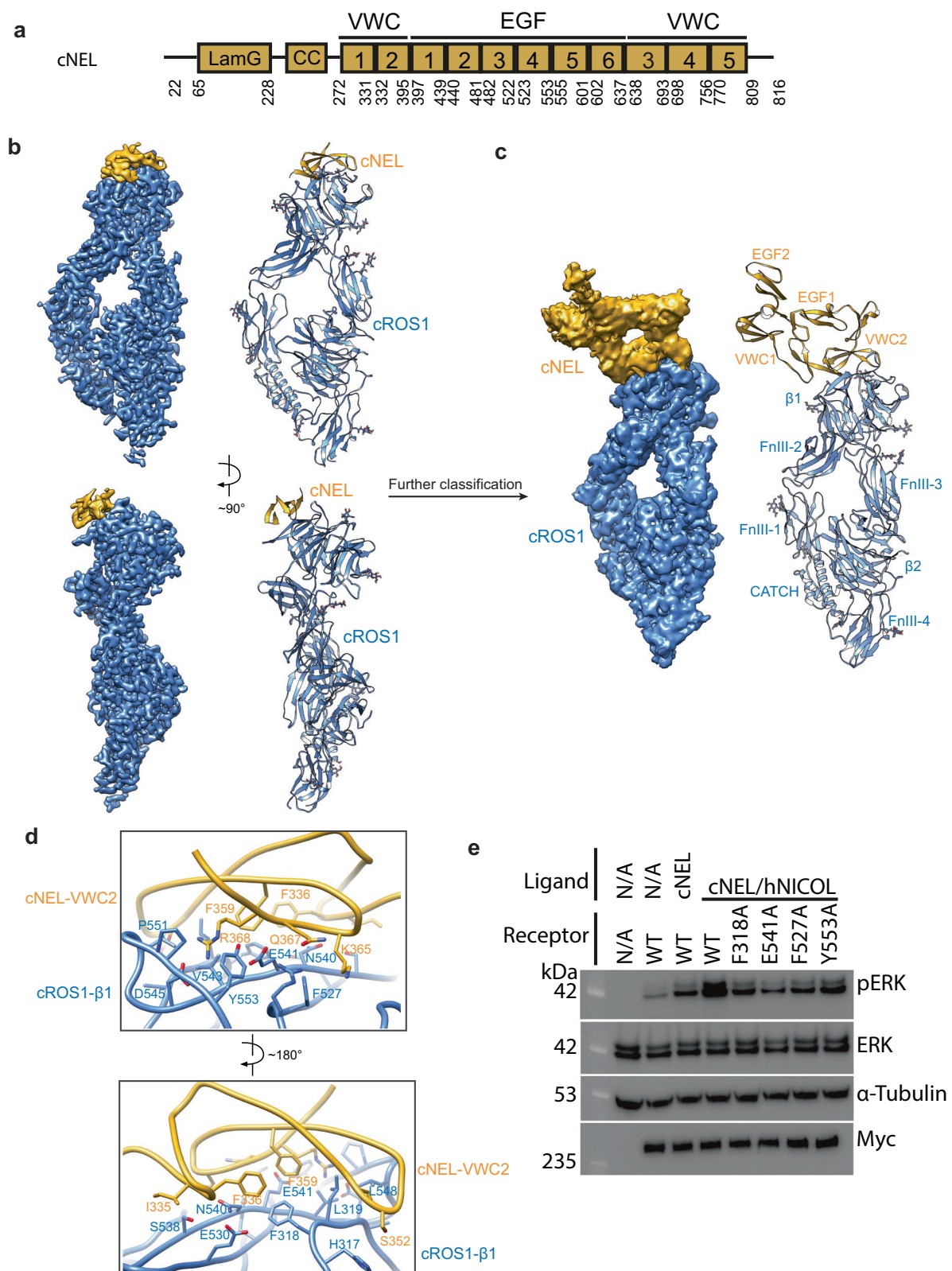

**Fig. 2 | The cryo-EM structure of cROS1/cNEL complex. a** Domain structures of chicken NEL. **b** 3D reconstruction of the cROS1/cNEL complex at 2.7 Å resolution and the corresponding ribbon representation of this complex. cROS1 and cNEL are shown in blue and yellow, respectively. Only the VWC2 domain of cNEL was resolved in the cryo-EM map. **c** After further classification, the VWC1, EGF1, and EGF2 domains of cNEL were resolved in the cryo-EM map of the cROS1/cNEL complex. **d** Close-up view of the interaction between the VWC2 domain of cNEL and

β1 domain of cROS1 shown in two views. cROS1 and cNEL are shown in blue and yellow, respectively. **e** cNEL or the cNEL/hNICOL complex induced ERK phosphorylation in HEK293 cells expressing full-length myc-tagged cROS1. ERK phosphorylation levels (pERK) were assessed by Western blot. Expression levels of cROS1 were monitored by anti-myc Western blot. The results shown are representative of three biological repeats.

or human NICOL (hNICOL), which shares 72% sequence identity (Fig. 3a and Supplementary Fig. 5b), in HEK293 cells. The cNEL/hNICOL complex exhibited higher expression levels and better biochemical properties compared to the cNEL/cNICOL complex (Supplementary Figs. 6 and 7a). We therefore chose the cNEL/hNICOL complex for further characterization. SDS-PAGE analysis revealed that cNEL and hNICOL form a highly stable complex, which could only be disrupted by reducing agents such as dithiothreitol (DTT), indicating that the complex is stabilized by disulfide bonds (Supplementary Fig. 6). The cNEL/hNICOL complex potently activates cROS1, leading to enhanced ROS1 autophosphorylation and downstream ERK phosphorylation (Supplementary Figs. 7b and 8). In addition, our cell-based assay showed that the cNEL/hNICOL complex activates ROS1 to a level comparable to that of the cNEL/cNICOL complex (Supplementary Fig. 7c), suggesting that the species-mixed complex does not introduce unintended artifacts and is valid for the mechanistic analyses.

The BLI results showed that cROS1/cNEL and cROS1/cNEL/hNICOL have nearly identical binding affinities (Supplementary Fig. 3). Nevertheless, consistent with previous findings[17], our cell-based assays demonstrated that the cNEL/hNICOL complex activates cROS1 much more robustly than cNEL alone (Fig. 2e). Moreover, dose-dependent ligand analyses showed that the cNEL/hNICOL complex induced robust cROS1 activation even at low concentrations, whereas cNEL alone failed to elicit strong activation, even at concentrations up to 2000 nM (Supplementary Fig. 9a). Time-course analysis further indicated that maximal cROS1 activation by either the cNEL/hNICOL complex or cNEL alone was reached at approximately 30 min, but the level of activation by cNEL alone was much lower (Supplementary Fig. 9b). These results suggest that cNEL alone is not optimal for cROS1 activation.

To elucidate the structural basis of NICOL's role in ROS1 activation, we determined the cryo-EM structure of the cROS1/cNEL/hNICOL complex (Supplementary Fig. 10). Initial cryo-EM analysis showed that, unlike the flexible state of cNEL in the cROS1/cNEL complex, cNEL forms a rigid asymmetric dimer in the presence of hNICOL, suggesting that hNICOL binding stabilizes the cNEL dimer (Fig. 3b).

Two distinct classes of the cROS1/cNEL/hNICOL complex were identified, each containing a single cROS1 molecule bound to either protomer 1 or protomer 2 of the cNEL dimer via the same VWC2–β1 interface observed in the cROS1/cNEL complex (Fig. 3b, c). The conformation of cROS1 in this complex remains virtually identical to ligand-free cROS1 and cROS1 in the cROS1/cNEL complex, confirming the notion that cNEL does not induce any conformational changes to cROS1. When the models of the two distinct classes are superimposed based on cNEL, the head region of the cROS1 molecule from one structure clashes severely with that from the second structure, explaining why only one cROS1 molecule can engage the asymmetric cNEL dimer at a time (Fig. 3c).

## The structure of 2:1 cNEL/hNICOL complex

After applying focused 3D refinement with density subtraction[22], the cryo-EM map of the cNEL/hNICOL complex was resolved at 4.1 Å resolution, allowing us to build a complete model for this complex (Fig. 3d, e). In this batwing shaped structure, the two cNEL protomers interact primarily through homotypic contacts between their CC domains (Fig. 3d). The cNEL dimer is further stabilized by a domain-swapping mechanism, in which the LamG domain of one protomer engages the other protomer (Fig. 4a, b).

Notably, only a single hNICOL molecule was observed in the cryo-EM map. The middle part of hNICOL adopts an α-helical conformation and forms a three-helix bundle with the CC domains of both cNEL protomers at the center of the batwing shaped complex (Fig. 4a, c). At each end of the hNICOL helix, three inter-molecular disulfide bonds link hNICOL to the CC domains of cNEL protomers 1 and 2, further reinforcing the complex (Fig. 4d). To validate this binding mode, we deleted the CC domain of cNEL, co-expressed and co-purified cNEL-

ΔCC mutant with hNICOL, and found that the cNEL-ΔCC mutant no longer binds to hNICOL (Fig. 4e; Supplementary Fig. 11).

In addition to the central hNICOL-helix/cNEL-CC interaction, the VWC5 domain from each cNEL protomer also engages both the hNICOL helix and the CC domain of cNEL. Specifically, the VWC5 domain of protomer 1 interacts with the C-terminal portion of the hNICOL helix and the CC domain of protomer 2, while the VWC5 domain of protomer 2 contacts the N-terminal portion of hNICOL and the CC domain of protomer 1 (Fig. 4c). These asymmetric interactions lead to distinct conformations of the two cNEL protomers (Fig. 3e), resulting in an overall asymmetric dimer configuration. Each cNEL protomer is further stabilized by distinct intra-molecular interactions, predominantly between their EGF and VWC domains.

Within this asymmetric dimer, the LamG domain of each cNEL protomer engages the VWC4 domain of the opposing protomer, albeit in distinct ways (Fig. 4b). On one wing of the asymmetric cNEL/hNICOL complex, a major β-sheet with a concave curvature in the LamG domain of protomer 1 forms tight contacts with the lateral surface of a three-stranded β-sheet in the VWC4 domain of protomer 2 (Fig. 4b). Additionally, a loop between β-strands in this LamG domain makes contacts with the VWC5 domain of protomer 2.

In contrast, on the other wing of the complex, the N-terminal tail of the LamG domain in protomer 2 contacts a surface formed by a β-sheet and a β-hairpin in the VWC4 domain of protomer 1 (Fig. 4b). Furthermore, the N-terminus of hNICOL adopts an extended loop conformation that interacts with both the LamG domain of protomer 2 and the VWC4 domain of protomer 1 (Fig. 4b, c), contributing to the stability of this LamG–VWC4 interface. The LamG domain of protomer 2 also directly interacts with the EGF5 domain of protomer 1 (Fig. 4b), providing an additional stabilizing interaction for the asymmetric cNEL dimer. Notably, because the LamG–VWC4 interactions observed in the cNEL/hNICOL complex do not involve hNICOL, we suspect that the same interactions may also occur in the NICOL-free cNEL dimer.

## Proposed activation mechanism of ROS1 induced by NEL and NICOL

As described above, the 2:1 cNEL/hNICOL complex can recruit only a single cROS1 molecule, due to a steric clash that would occur if both protomers simultaneously bound cROS1. Since RTK activation generally requires receptor dimerization or higher-order oligomerization, this 1:2:1 cROS1/cNEL/hNICOL configuration alone cannot explain how NEL and NICOL mediate ROS1 activation. This structural observation suggests that the 1:2:1 complex must engage in higher-order assembly to enable cROS1 activation. Indeed, both SEC-MALS and blue native PAGE (BN-PAGE) results showed that cROS1 alone predominantly exists as a monomer, whereas the cNEL, cNEL/hNICOL, cROS1/cNEL, and cROS1/cNEL/hNICOL complexes can further assemble into higher-order oligomers (Fig. 5a, b).

Supporting this idea, low-pass-filtered cryo-EM maps of both classes of the cROS1/cNEL/hNICOL complex reveal additional density adjacent to the VWC4 domain of protomer 2 that is consistent in size and shape with a LamG domain, suggesting the presence of a second hetero-tetrameric complex (Supplementary Fig. 12). Based on these observations, we propose that a 1:2:1 cROS1/cNEL/hNICOL complex may associate with a second heterotetrametric complex through an interaction between the VWC4 domain from the first complex and the LamG domain from the second complex (Supplementary Fig. 12). Through the same interaction, the cROS1/cNEL/hNICOL complex could form daisy chain-like higher-order oligomers, in which multiple ROS1 molecules can promote their trans-autophosphorylation and activation.

To test this model, we expressed and purified a LamG-deleted version of cNEL, designated cNEL-ΔLamG. SDS-PAGE and pull-down assays demonstrated that cNEL-ΔLamG could still associate with hNICOL (Fig. 4e and Supplementary Fig. 11) and bind cROS1 with the same

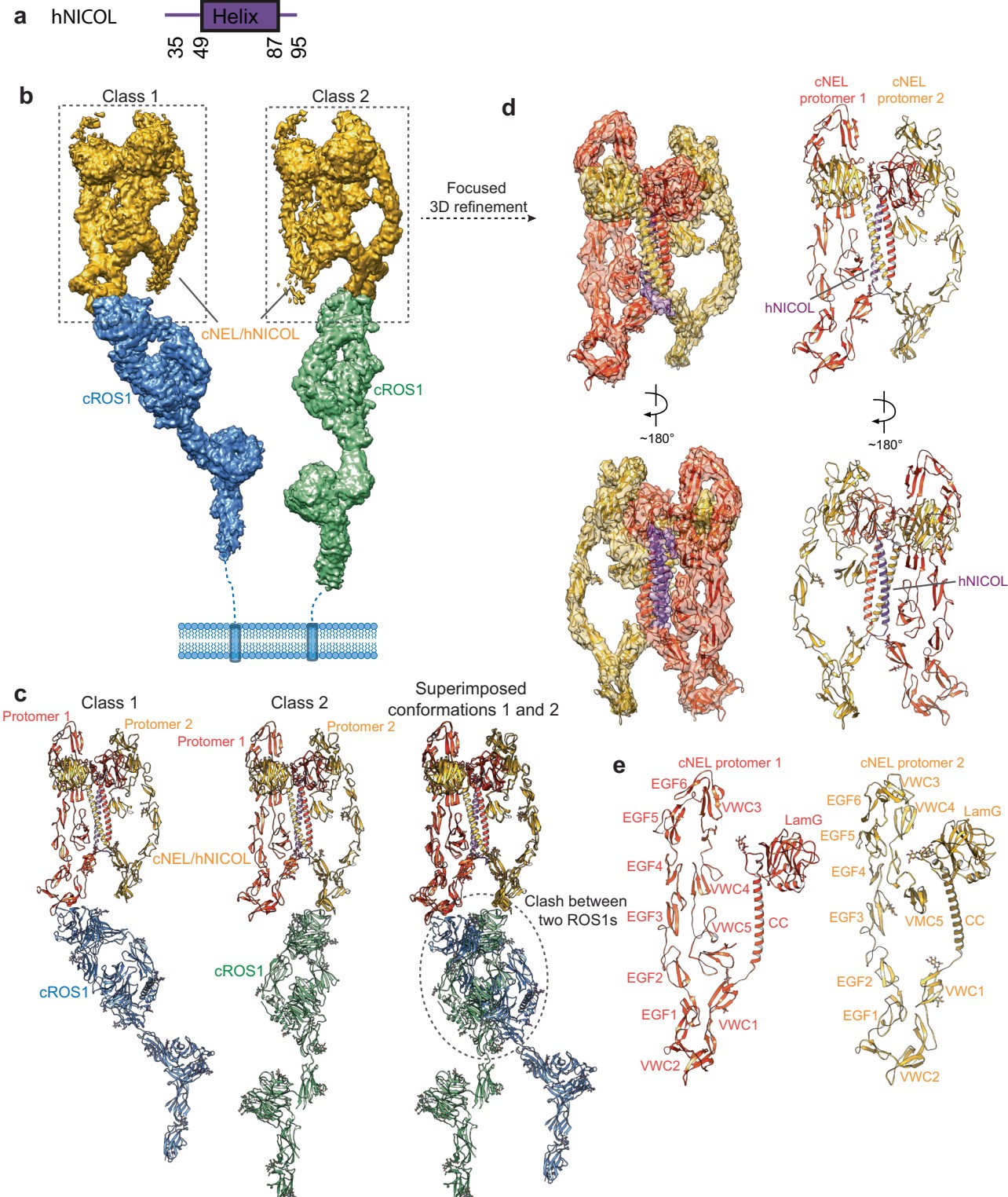

**Fig. 3 | Overall structure of 1:2:1 cROS1/cNEL/hNICOL complex. a** Domain structures of human NICOL **b** Cryo-EM maps of two distinct 1:2:1 cROS1/cNEL/hNICOL complexes. In class 1, one cROS1 is bound at protomer 1 of cNEL. In class 2, one cROS1 is bound at protomer 2 of cNEL. The cryo-EM density corresponding to cNEL/hNICOL is colored in yellow. The cryo-EM density corresponding to cROS1 is colored in blue (class 1) or green (class 2). **c** The ribbon representations of two distinct 1:2:1 cROS1/cNEL/hNICOL complexes. One cROS1 is bound at either protomer 1 of cNEL (class 1) or protomer 2 of cNEL (class 2). The superimposition of classes 1 and 2 reveals steric clash between protomers 1 and 2 bound cROS1,

explaining why the 2:1 cNEL/hNICOL complex can only recruit one cROS1. **d** 3D reconstruction of the 2:1 cNEL/hNICOL complex after focused 3D refinement on the top region of the 1:2:1 cROS1/cNEL/hNICOL complex, and the corresponding ribbon representation of this complex fitted into the cryo-EM maps shown in two views. The cryo-EM map was resolved at 4.1 Å resolution. Two cNEL and one hNICOL molecules were identified and built. hNICOL is bound at the CC regions of the cNEL dimer. Protomers 1 and 2 of cNEL are colored in orange and yellow, respectively. hNICOL is colored in purple. **e** Ribbon representations of individual protomers 1 and 2 of cNEL, highlighting their distinct conformations.

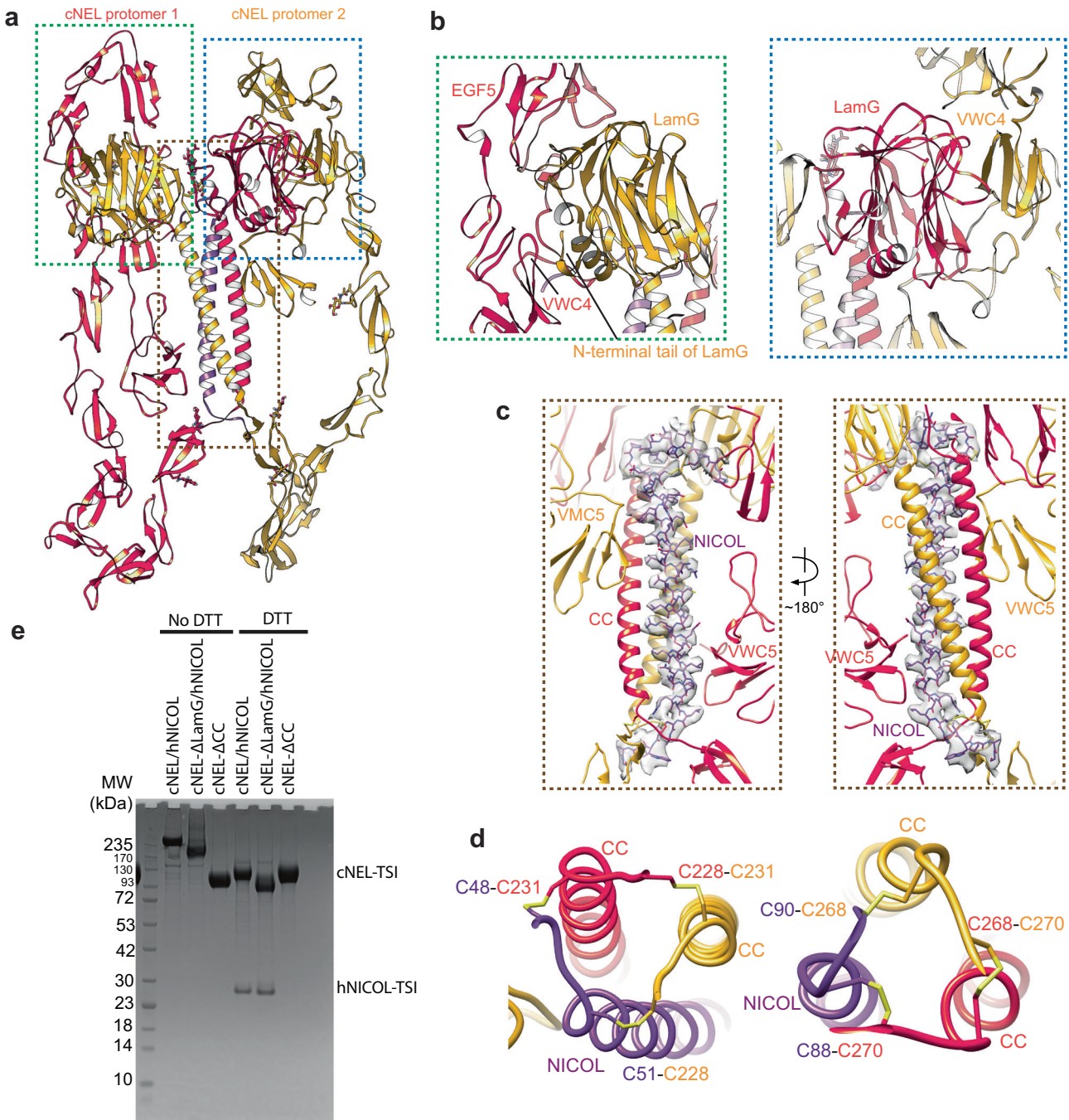

**Fig. 4 | The structure of 2:1 cNEL/hNICOL complex. a** The overall structure of 2:1 cNEL/hNICOL complex. Protomers 1 and 2 of cNEL are colored in orange and yellow, respectively. hNICOL is colored in purple. **b** Green box: close-up view of the interaction between LamG domain of cNEL protomer 2 and EGF5 and VWC4 domains of cNEL protomer 1. Blue box: close-up view of the interaction between LamG domain of cNEL protomer 1 and VWC4 domain of cNEL protomer 2. The locations of these interactions are indicated by color-matched boxes in (**a**). **c** Close-up view of the interaction between CC domains of the cNEL dimer and hNICOL. The CC domains of cNEL are colored in orange and yellow, respectively. hNICOL is shown as stick model and colored in purple. The cryo-EM density corresponding to the hNICOL is shown. The location of this interaction is indicated by a brown box in (**a**). **d** Close-up view of the disulfide bridges between CC domains of cNEL dimer and hNICOL. A total of 6 disulfide bridges are formed between CC domains of the cNEL dimer and hNICOL. **e** SDS-PAGE analysis of co-expressed and co-purified cNEL/hNICOL samples following size exclusion chromatography (SEC) reveals that hNICOL forms a covalently linked complex with cNEL. Deletion of the CC domain in hNICOL–but not the LamG domain–disrupts this complex. The results shown are representative of two replicates.

affinity as the wild-type cNEL/hNICOL complex (Supplementary Fig. 13a). We next assessed whether cNEL/hNICOL can self-assemble into higher-order oligomers using BN-PAGE. Consistent with our structural model, the wild-type cNEL/hNICOL complex formed tetramers, hexamers, and larger oligomeric species, whereas the cNEL-ΔLamG/hNICOL complex failed to form higher-order assemblies (Fig. 5b). Additionally, the cNEL-ΔCC mutant predominantly existed in

a monomeric state, with only a small portion forming high-order oligomer (Fig. 5b). In cell-based assays, the wild-type cNEL/hNICOL complex robustly activated cROS1, whereas both the cNEL-ΔLamG/hNICOL and cNEL-ΔCC mutants exhibited remarkably reduced ability on inducing cROS1 activation (Fig. 5c). Furthermore, we titrated a cNEL mutant lacking both the Coiled-Coil and LamG domains (cNEL-ΔCC-ΔLamG), which contains the VWC2 domain required for strong cROS1

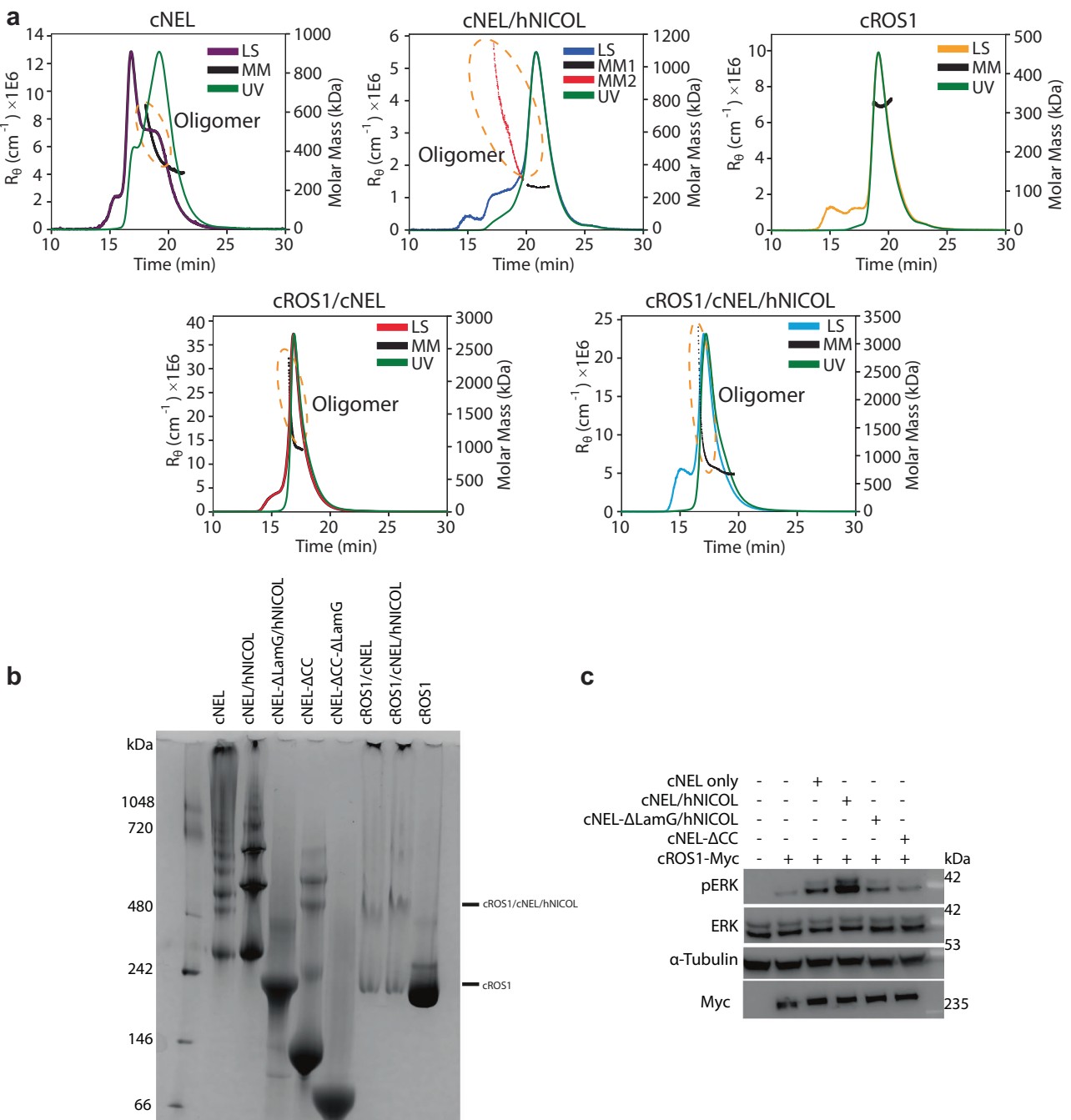

**Fig. 5 | cROS/cNEL/hNICOL complexes further assemble into high-order oligomers. a** SEC-MALS profile of cNEL, cNEL/hNICOL, cROS1, cROS1/cNEL, and cROS1/cNEL/hNICOL samples. Solid lines represent the light scattering in physical units (left axis), and markers show the respective molar masses (right axis). Green line represents UV trace. The UV readings are normalized such that the highest UV reading is equal to the highest LS reading. For the cNEL/hNICOL sample, due to the extensive mass range of the oligomers, the masses from the main species (black; MM1) have been separated from the oligomers (red; MM2). The corresponding UV trace is shown to illustrate the relative population of these species on an extinction basis. **b** The ability of different ligands or ligand/receptor complexes to form higher-order oligomers was assessed using blue native PAGE (BN-PAGE). All cNEL, cNEL/hNICOL, and cNEL-ΔCC are capable of forming oligomers. However, as compared with cNEL and cNEL/hNICOL, cNEL-ΔCC shows a remarkably reduced ability to oligomerize. cNEL-ΔLamG/hNICOL and cNEL-ΔCC-ΔLamG are not able to form higher-order oligomers. cROS1 alone primarily exists as a monomer, whereas both cROS1/cNEL and cROS1/cNEL/hNICOL can effectively form large higher-order oligomers. Notably, some of the cROS1/cNEL or cROS1/cNEL/hNICOL samples did not enter the gel due to their large size. The results shown are representative of two replicates. **c** cNEL, cNEL/hNICOL, and various truncation mutants were tested for their ability to induce ERK phosphorylation in HEK293 cells expressing full-length myc-tagged cROS1. ERK phosphorylation (pERK) was assessed by western blot, and cROS1 expression levels were monitored using an anti-myc antibody. Deletion of the LamG or CC domains in cNEL abolished its ability to activate cROS1. The results shown are representative of three biological replicates.

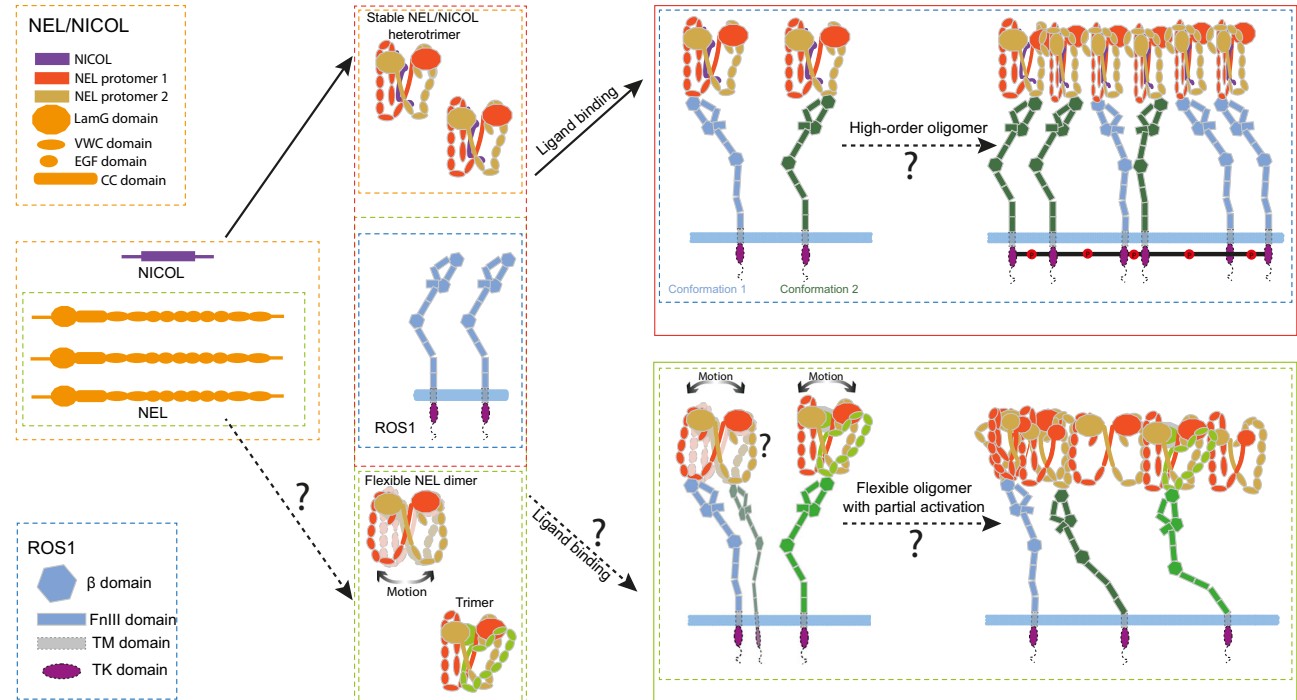

**Fig. 6 | Cartoon representation of a working model for NEL and NICOL dependent activation of ROS1.** NEL are secreted and assembled into stable dimer in the presence of its co-ligand NICOL. Excess NEL are assembled into flexible dimer or trimmer in the absence of NICOL. Free NEL/NICOL co-ligands or NEL-only ligands are secreted and enter the circulation to reach the target cell surface. NEL/NICOL engages ROS1 through a strong interaction between the VWC2 domain of cNEL and the β1 domain of cROS1. NEL/NICOL complexes further assemble into higher-order oligomers through the interaction between the VWC4 domain of NEL protomer 2 in one complex and the N-terminal tail of the LamG domain of NEL protomer 1 in another. Formation of these higher-order ligand oligomers drives ROS1 clustering and triggers downstream signaling. Flexible NEL dimer or trimmer can also recruit ROS1 through the strong interaction between the VWC2 domain of cNEL and the β1 domain of ROS1. In contrast to NEL/NICOL, NEL dimer and trimmer form dynamic oligomers that are less potent in activating ROS1. NEL protomers 1 and 2 are shown in red and orange, respectively. Distinct domains of both ligand and receptor are depicted using different shapes. TM domain: transmembrane domain; TK domain: tyrosine kinase domain. The molecular processes supported by structural data are indicated in solid arrows, and the processes not directly supported by structural data are indicated in dashed arrows and "?".

binding but cannot induce cROS1 activation (Supplementary Fig. 13b), into wild-type cNEL/hNICOL-dependent cROS1 activation assays. The results showed that cNEL-ΔCC-ΔLamG inhibited WT cNEL/hNICOL−induced cROS1 activation in a dose-dependent manner (Supplementary Fig. 13c). These findings suggest that both the binding of cNEL/hNICOL to cROS1 and the ligand indued higher-order oligomerization are required for cROS1 activation.

## Discussion

Based on our structural analyses, we propose a model for how NEL and NICOL drive ROS1 activation. In the ligand-free state, ROS1 exists as a monomer and adopts an arc-shaped conformation. NEL alone is structurally flexible but can bind one ROS1 molecule via an interaction between its VWC2 domain and the β1 domain of ROS1. Importantly, ROS1 does not undergo significant conformational changes upon binding NEL. One NICOL molecule binds to the dimeric CC domains of NEL, forming a three-helix bundle stabilized by six inter-molecular disulfide bonds at both ends of the helices. This binding event rigidifies the NEL dimer into a domain-swapped, asymmetric conformation, resulting in a 2:1 NEL/NICOL complex with a batwing-like architecture. Each "wing" of the complex is stabilized by a network of distinct intra- and inter-molecular interactions: EGF–VWC interactions within individual NEL protomers, and LamG–VWC4 and VWC5−NICOL/CC interactions between protomers. Crucially, the 2:1 NEL/NICOL complex can recruit only a single ROS1 molecule−either via the VWC2 domain of cNEL protomer 1 or protomer 2−due to a steric clash that occurs when two ROS1 molecules are to bind simultaneously. Nevertheless, we cannot rule out the possibility that a 2:1 NEL/NICOL complex could recruit two ROS1 molecules following significant conformational changes in either NEL/NICOL or ROS1.

Our structural data further reveal that these 2:1 NEL/NICOL complexes can oligomerize into higher-order assemblies. We propose that this oligomerization may occur through interactions between the VWC4 domain of NEL protomer 2 from one heterotetrameric complex and the N-terminal tail of the LamG domain of NEL protomer 1 from another heterotetrameric complex. These higher-order assemblies bring multiple ROS1 molecules into proximity, facilitating trans-autophosphorylation and activation of downstream signaling pathways (Fig. 6). Because the LamG-N-tail/VWC4 interaction is relatively weak and no direct contacts are formed between the ROS1 molecules themselves, the resulting ROS1/NEL/NICOL supercomplex remains flexible, which explains why only one complete 1:2:1 ROS1/NEL/NICOL complex could be resolved in our cryo-EM analysis. Further structural studies will be required to elucidate the molecular details underlying the assembly of higher-order ROS1/NEL/NICOL complexes.

Our SDS-PAGE, BN-PAGE, and SEC-MALS results showed that NEL alone primarily exists as a disulfide-linked dimer and can also further oligomerize even in the absence of NICOL, presumably through similar LamG-VWC4 interaction. Our biochemical and cryo-EM data indicate that NEL exhibits markedly improved biochemical stability and reduced aggregation upon NICOL binding, suggesting that NICOL may act as a chaperon to enhance the biological activity of NEL by stabilizing its structure and improving its solubility. In addition to the chaperone effect, NICOL may facilitate NEL-dependent ROS1 activation through enhancing the formation of the ROS1 active oligomer and positioning multiple ROS1 molecules in a more favorable configuration

for activation, although the details of this mechanism remain unclear (Fig. 6). Interestingly, NEL and NICOL display highly similar expression patterns across tissues (based on the Human Protein Atlas). In particular, both NEL and NICOL are highly expressed in the brain and male reproductive tissues. This raises the intriguing possibility that NEL is always associated with NICOL in vivo and that NICOL functions as a constitutive structural partner of NEL. Further work is required to answer these questions.

Ligand-induced dimerization is a common mechanism for RTK activation. In this classical model, the ligand itself forms a dimer and brings two receptor molecules into close proximity, thereby facilitating cross-phosphorylation and activation[2]. In contrast, our findings reveal a distinct mechanism for ROS1 activation. Although NEL/NICOL forms a heterotrimer, it can only recruit a single ROS1 molecule due to steric hindrance, and thus is incapable of activating ROS1. Instead, ROS1 activation requires the higher-order oligomerization of the NEL/NICOL heterotrimer, which enables the clustering of multiple ROS1 molecules. This ligand multimerization-driven activation mechanism represents a special paradigm among RTKs; however, the precise activation mechanism of ROS1 remains unresolved and will require further investigation.

Additionally, it has been known that several RTKs—such as RET[23,24] and MuSK[25,26]—require co-receptors for activation, while others, such as FGFR[27] and c-MET[28], require heparin to achieve full activation. Intriguingly, ROS1 is an RTK that depends on a co-ligand, NICOL, for activation. NICOL does not directly bind ROS1; rather, it covalently associates with the NEL dimer to form a rigid, asymmetric 2:1 NEL/NICOL heterotrimeric complex. This stabilized conformation is optimal for supporting ROS1 activation. Interestingly, NEL has also been shown to activate other receptors, such as ROBO3[21,29]. However, it remains unclear whether NICOL is also required for NEL-dependent activation of ROBO3. Further investigation will be needed to determine whether this co-ligand mechanism extends beyond ROS1.

In conclusion, our structural and functional analyses provide comprehensive insights into the molecular mechanism by which the NEL/NICOL complex activates ROS1. These findings not only elucidate a distinct mode of receptor tyrosine kinase activation but also pave the way for developing potential therapeutic strategies targeting diseases driven by aberrant ROS1 signaling, including cancer and male infertility.

## Methods

### Oligonucleotides and plasmids
Detailed oligonucleotides and plasmid information is provided within the Source Data File. All of the primers were synthesized by Sigma Aldrich. The cDNA templates for cROS1 and cNEL were purchased from GenScript. The ORF sequences for cNICOL and hNICOL amplification were synthesized by IDT. All the constructs were made according to the directions described in "Method" section.

### Chicken ROS1 expression and purification
The DNA sequence encoding the full-length extracellular domain of cROS1 (residues 1–1873) was amplified by polymerase chain reaction (PCR) from a cDNA template (GenScript, Oga00025) and cloned into our previously modified pEZT-BM vector, which contains an H3C cleavage site followed by Tsi and His tags at the C-terminus, using the Gibson assembly method. The amino-acid sequences of linkers/H3C cleavage site/tags appended to the C-terminus is "GTSSG-LEVLFQG P-VDFDKTLTHPNGLVVERPVGFDARRSAEGFRFDEGGKLRNPRQLEVQR QDAPPPPDLASRRLGDGEARYKVEEDDGGSAGSEYRLWAAKPAGARWIV VSASEQSEDGEPTFALAWALLERARLQ-SS-HHHHHHHH".

The expression and purification of cROS1 were performed as described in previous study with some modifications[30]. In brief, pEZT-BM-cROS1-H3C-Tsi-His8 plasmid was transformed into the DH10Bac

bacteria to generate bacmid DNA. The bacmid DNA was introduced into Sf9 (Gibco, 11496015) cell cultures in 6-well culture plates (1 million per well) with 2 ml Insect-XPRESSTM Medium (Lonza, 12730Q) at 27 °C to produce recombinant baculovirus by using Cell-fectin reagent (20 μg bacmid DNA: 8 μL Cellfectin™ II Reagent (10362100)). Twelve hours after transfection, 1% antibiotics and 2% FBS were added. Six days after transfection, collect cell culture medium containing P1 virus. P2 viruses were produced by adding 0.5 ml P1 containing medium into 50 ml Sf9 cell culture (cell density around 2.5 million per ml). Five days after infection, collect cell culture medium containing P2 virus. 3 ml of P2 virus containing medium add into 300 ml Sf9 cells culture to produce P3 virus. Four days after P2 infection, collect P3 virus to express target proteins.

Protein was expressed in FreeStyle 293-F (Invitrogen R79007) cells cultured in FreeStyle™ 293 Expression Medium (Gibco, 12338-018) by infecting the virus at a ratio of 3:100 (virus: cell, v/v). The infected cells were supplemented with 10 mM sodium butyrate to boost protein expression 12 h after virus infection. The cells were cultured at 37 °C and 8% $CO_2$.

The cell culture medium was harvested 96 h after sodium butyrate supplement and filtered. $CaCl_2$ was added to the filtered cROS1-containing cell culture medium to a final concentration of 5 mM prior to incubation with pre-equilibrated $Ni^{2+}$-NTA (nickel) resin(CuboBio-tech, 75105) at 4 °C for 6–8 h. The cell culture medium, together with resin, was loaded into a column and drained under gravity. The nickel resin was washed with washing buffer 1 (20 mM Tris-HCl, pH 8.0, 150 mM NaCl, 3 mM $CaCl_2$, 20 mM imidazole) by gravity flow for 2 times and then washed with washed with wash buffer 2 (20 mM Tris-HCl, pH 8.0, 150 mM NaCl, 3 mM $CaCl_2$, 40 mM imidazole). The resin was eluted with elution buffer (20 mM Tris-HCl, pH 8.0, 150 mM NaCl, 250 mM imidazole). The eluted protein was concentrated and cleavage by HRV-3C protease to remove the tag. The protein sample was then run on Superose 6 increase 10/300 GL size-exclusion column with GF buffer (20 mM HEPES-Na buffer pH 7.5; 150 mM NaCl and 2 mM $CaCl_2$). The peak fractions were combined, concentrated, flash frozen with liquid nitrogen, and then stored at −80 °C for subsequent experiments.

### Chicken NEL expression and purification
DNA sequence encoding the full-length of cNEL was amplified by Polymerase Chain Reaction (PCR) from cDNA sequence (GenScript, Oga27602) and cloned into our previously modified pEZT-BM vector containing a H3C cleavage site followed by Tsi and His tag on C-terminal. Mutant cNEL-expressing plasmids were generated from the pEZT-BM-cNEL-H3C-Tsi-His8 vector using Q5 site-directed mutagenesis. For cNEL-ΔLamG mutant, the sequence before CC domain(28-212) was deleted. The deleted sequence is LQIDVLSELGLPG YAAGVRQVPGLHNGSKAFLFPDTSRSVKASPETAEIFFQKLRNKYEFTILVT LKQ AHLNSGVIFSIHHLDHRYLELESSGHRNEIRLHYRTGSHRSHTEVFPYI LADDKWHRLSLAISASHLILHVDCNKIYERVVEKPFMDLPVGTTFWLGQR NNAHGYFKGIMQDVQLLV. For cNEL-ΔCC mutant, the sequence around CC domain (228-270) CPTCNDFHGLVQKIMELQDILAKTSAKL SQAEQRMNKLDQCYC was replaced by linker sequence SPTSGSGS SGGSSGGSSGGSSGGSDQSYS.

For cNEL-ΔCC-ΔLamG mutant, the sequence around CC domain and before CC domain (28-270) was deleted. The deleted sequence is LQIDVLSELGLPGYAAGVRQVPGLHNGSKAFLFPDTSRSVKASPETAEIFFQ KLRNKYEFTILVTLKQAHLNSGVIFSIHHLDHRYLELESSGHRNEIRLHYRT GSHRSHTEVFPYILADDKWHRLSLAISASHLILHVDCNKIYERVVEKPFMDL PVGTTFWLGQRNNAHGYFKGIMQDVQLLVMPQGFISQCPDLNRTCPTC NDFHGLVQKIMELQDILAKTSAKLSQAEQRMNKLDQCYC.

The baculovirus production, protein expression, and purification procedures for cNEL were the same as those used for cROS1, as described above.

## Co-expression and co-purification of cNEL/hNICOL complex

DNA sequence encoding human NICOL and chicken NICOL were synthesized by Integrated DNA Technologies (IDT, Inc.), and cloned into our previously modified pEZT-BM with Gibson assembly method to construct pEZT-BM-hNICOL-H3C-Tsi-His8 expression vector.

The baculovirus production for cNEL and hNICOL was the same as that used for cROS1, as described above. To isolate cNEL/hNICOL protein complex, we applied co-expression approach by adding cNEL and hNICOL expression baculovirus into Freestyle 293-F cells at the same time. The cNEL and hNICOL viruses were added at a ratio of 1:2(20 ml cNEL -encoding P3 virus and 40 ml hNICOL-encoding P3 virus were added to 1 L Freestyle 293-F cell culture) to ensure excess expression of hNICOL. The following procedures for expression and purification of cNEL/hNICOL were totally the same as those used for cROS1.

## cROS1/cNEL or cROS1/cNEL/hNICOL complex reconstitution

Gel filtration purified cNEL or cNEL/hNICOL was mixed with cROS1 at molar ratio of 1:1 and incubated on ice for 30 min to reconstitute ligand-receptor complex. The sample was then run on a Superose 6 increase 10/300 GL size-exclusion column with GF buffer (20 mM HEPES-Na buffer pH 7.5; 150 mM NaCl and 2 mM $CaCl_2$). The peak fractions were collected and concentrated using an Amicon Ultra concentrator with a 100 kDa cut-off (Millipore) to different concentration accordingly.

## ROS1 signaling assay

For ROS1 signaling assay, the DNA sequence encoding the full-length of cROS1 was amplified by polymerase chain reaction (PCR) from a cDNA template (GenScript, Oga00025) and cloned into modified pcDNA3.1(-) vector, which contains an EGFP tag followed by a myc tag at the C-terminus, using the Gibson assembly method. Point mutant ROS1-expressing plasmids were generated from the pcDNA3.1-cROS1-EGFP-myc vector using Q5 site-directed mutagenesis.

For phosphorylation of ERK assessment, Vigorous 293FT (Invitrogen, R70007) cells were cultured in DMEM medium supplemented with 10% FBS and 1% antibiotics under standard cell culture conditions. Subculture the cells into 6-well culture plates and seed approximately 0.8 million cells per well when they reach 80% confluency. After 1 day, plasmid transfection was performed using PEI at a 1:3 (DNA:PEI) mass ratio to express EGFP-Myc-tagged cROS1 and its mutants in 293FT cells. One day after plasmid transfection, most cells exhibited green fluorescence. The cells were then serum-starved for 10 h. Serum-starved cells were treated for 25 minutes with cNEL, cNEL/hNICOL, or mutant ligands, each diluted in high-glucose DMEM without serum to a final concentration of 100 nM. After treatment, cells were incubated with cell lysis buffer [20 mM Hepes 7.5; 150 mM NaCl; 1% triton X100] supplemented with cOmplete Protease Inhibitor Cocktail (Roche) on ice for 10 min. After centrifugation at 20,600 g at 4 °C for 15 min, cell lysate supernatant samples were made with 4X SDS DTT containing buffer. The samples were then separated by SDS-PAGE and analyzed by Western blotting. anti-Myc (1:2000; Cell Signaling,2278), anti-pERK1/2 (1:2000; Cell Signaling,4370), anti-ERK1/2 (1:2000; Cell Signaling, 4695), and anti-alpha-tubulin (1:2000; Cell Signaling, 3873) were used as primary antibodies. Anti-rabbit IgG, HRP-linked Antibody (Cell Signaling, 7074) and Anti-mouse IgG, HRP-linked Antibody (Cell Signaling, 7076) were used as secondary antibodies accordingly.

For phosphorylation assessment of cROS1, NCI-H1299(ATCC, CRL-5803 ™) cells were cultured in RPMI-1640 medium (Gibco, 11875093) supplemented with 10% FBS and 1% antibiotics under standard cell culture conditions. Cells were subcultured into 6-well plates and seeded at approximately $1 \times 10^6$ cells per well. After 2 days, when the cells reached ~80–87% confluency, plasmid transfection was performed using Lipofectamine™ 2000 to express EGFP-Myc-tagged cROS1. 12 h after transfection, the medium was replaced with fresh

growth medium. Two days after transfection, the cells were serum-starved for 16 h. Serum-starved cells were then treated with cNEL/hNICOL (100 nM, diluted in serum-free RPMI-1640) for 25 min. Subsequent cell lysis and western blot procedures were performed as described above for 293FT cells. Anti-pROS1 (Tyr2274)(1:1000; Cell Signaling, 3078) was used as primary antibodies.

## Statistics and reproducibility

ImageJ 1.53e was used to quantify band intensities from Western blot experiments. GraphPad Prism 5 was used to generate graphs and perform statistical analyses. Data are presented as mean ± s.d. from $n = 3$ independent experiments. A two-tailed Student's $t$-test was used for comparisons between the ligand-site–mutated group and the no-ligand–treated group. No statistical method was used to predetermine sample size. No data were excluded from the analyses. Randomization and blinding methods were not used.

## Pull-down assay

For samples involving cROS1/cNEL interface mutants, 10 μg of bait protein (cNEL-TSI3-His or TSI tagged mutants) and 10 μg of prey protein (cROS1) were mixed in 500 μl of TBS buffer (20 mM Tris-Cl, pH 7.5; 150 mM NaCl; 0.1% Tween-20) supplemented with 2 mM $CaCl_2$, and incubated on ice for 1 h. Pre-equilibrated TSE resin was then added, and the mixture was incubated overnight at 4 °C with gentle rotation. After incubation, the samples were centrifuged at 400 × $g$ for 2 min, and the supernatant was discarded. The resin was washed three times with TBS buffer containing 2 mM $CaCl_2$ to remove unbound proteins. Bound proteins were eluted by boiling the resin in SDS loading buffer supplemented with DTT, and analyzed by SDS–PAGE.

## Native gel analyses cNEL/hNICOL oligomerization

Native gel electrophoresis was performed as previously described by our laboratory[31]. Briefly, purified cNEL/hNICOL wild-type or mutant protein complexes (15 μg) were prepared in GF buffer (20 mM HEPES-Na, pH 7.5; 150 mM NaCl; 2 mM $CaCl_2$), mixed with NativePAGE™ sample buffer (Invitrogen, Cat. No. BN20032), and resolved on a 3–12% Bis-Tris gradient native gel (Invitrogen, Cat. No. BN2012BX10).

## SEC-MALS experiments

All size-exclusion chromatographic multi-angle light-scattering (SEC-MALS) experiments were conducted at 4 °C. A Superdex 200 Increase GL 10/300 column, attached to an ÄKTA pure system (GE), was equilibrated with MALS buffer (20 mM HEPES, pH 7.5 150 mM NaCl, and 2 mM $CaCl_2$). Protein samples (200 μL; filtered through a 0.22 μm centrifugal filter) at 1 mg/mL were injected onto the column and detected with the ÄKTA's UV detector and a miniDAWN TREOS detector (Wyatt Technologies). The flow rate was 0.5 mL/min. Data were analyzed using ASTRA version 7.1.0.29 (Wyatt). Concentration information was gleaned from the UV absorbance and the respective calculated mass-based extinction coefficients. For protein complexes, these values were surmised based on relative elution volumes and likely molar ratios. A sample of bovine serum albumen (Pierce) was used to calibrate the apparatus. Reported molar masses are the weight-averages of the masses observed across the dominant light-scattering peak. In some cases, it was necessary to exclude strongly sloping and/or noisy parts of the molar-mass distribution in the calculation of these weight-averages.

## BLI assay

Binding affinity of ROS1 to NEL or NEL/NICOL was determined by BLI using the Octet® R2 system (Sartorius). The cROS1 protein was biotinylated using EZ-Link NHS-PEG4-Biotin (ThermoFisher Scientific, MAN0016360) at a protein-to-biotin molar ratio of 1:15. Excess biotin was removed by gel filtration. 5 nM of biotinylated cROS1 in kinetic buffer (20 mM Hepes, 150 mM NaCl, and 2 mM $CaCl_2$ at pH 7.5, 0.1% (w/v) BSA, and 0.02% Tween-20) were captured on Streptavidin (SA)

**Table 1 | Cryo-EM data collection and refinement statistics**

| | Ligand-free-ROS1 head region | Ligand-free-ROS1 leg region | Ligand-free-ROS1 | ROS1/NEL | NEL/NICOL | ROS1/NEL/NICOL | ROS1/NEL/NICOL class 1 | ROS1/NEL/NICOL class 2 |
|---|---|---|---|---|---|---|---|---|
| **Data collection and processing** | | | | | | | | |
| Magnification | 105,000 | 105,000 | 105,000 | 105,000 | 105,000 | 105,000 | 105,000 | 105,000 |
| Voltage (kV) | 300 | 300 | 300 | 300 | 300 | 300 | 300 | 300 |
| Electron exposure (e/Å²) | 60 | 60 | 60 | 60 | 60 | 60 | 60 | 60 |
| Defocus range (μm) | 1.2–2.2 | 1.2–2.2 | 1.2–2.2 | 1.2–2.2 | 1.2–2.2 | 1.2–2.2 | 1.2 – 2.2 | 1.2 – 2.2 |
| Pixel size (Å) | 1.18 | 1.18 | 1.18 | 0.936 | 1.4042 | 1.4042 | 1.4042 | 1.4042 |
| Symmetry imposed | C1 | C1 | C1 | C1 | C1 | C1 | C1 | C1 |
| Initial particle images (no.) | 4,055,236 | 4,055,236 | 4,055,236 | 4,154,901 | 3,711,182 | 3,711,182 | 3,711,182 | 3,711,182 |
| Final particle images (no.) | 255,602 | 255,602 | 94,528 | 141,298 | 33,485 | 17,461 | 16,841 | |
| Map resolution (Å) | 2.9 | 3.6 | 3.4 | 2.7 | 4.1 | 4 | 4.2 | |
| FSC threshold | 0.143 | 0.143 | 0.143 | 0.143 | 0.143 | 0.143 | 0.143 | 0.143 |
| **Refinement** | | | | | | | | |
| **Model composition** | | | | | | | | |
| Nonhydrogen atoms | 8248 | 6522 | 14787 | 8654 | 12814 | 26142 | 26076 | |
| Protein residues | 1005 | 780 | 1788 | 1054 | 1638 | 3254 | 3243 | |
| Ligands | 15 N-acetylglucosamines | 18 N-acetylglucosamines, 1 β-D-mannose | 33 N-acetylglucosamines, 1 β-D-mannose | 9 N-acetylglucosamines | 9 N-acetylglucosamines | 36 N-acetylglucosamines | 36 N-acetylglucosamines | |
| **R.m.s. deviations** | | | | | | | | |
| Bond lengths (Å) | 0.002 | 0.004 | 0.002 | 0.004 | 0.003 | 0.002 | 0.002 | |
| Bond angles (°) | 0.525 | 0.575 | 0.529 | 0.537 | 0.572 | 0.621 | 0.588 | |
| **Validation** | | | | | | | | |
| MolProbity score | 1.10 | 1.42 | 1.33 | 1.38 | 1.37 | 1.47 | 1.63 | |
| Clashscore | 1.30 | 3.32 | 2.30 | 3.05 | 2.11 | 2.56 | 2.58 | |
| Poor rotamers (%) | 0 | 0.28 | 0.25 | 0.21 | 0.75 | 1.51 | 1.75 | |
| **Ramachandran plot** | | | | | | | | |
| Favored (%) | 96.29 | 95.70 | 95.42 | 95.89 | 94.29 | 95.76 | 94.07 | |
| Allowed (%) | 3.71 | 4.30 | 4.58 | 4.11 | 5.71 | 4.18 | 5.90 | |
| Disallowed (%) | 0 | 0 | 0 | 0 | 0 | 0.06 | 0.03 | |

Biosensors (Sartorius, 18-0009) for 120 s, followed by a baseline measurement in kinetic buffer for 180 s. cNEL/hNICOL(containing 3.13 nM, 6.25 nM, 12.5 nM, 25 nM and 50 nM) or cNEL (containing 0.41 nM, 1.23 nM, 3.7 nM, 11.1 nm and 33.3 nM) were associated to biotinylated ROS1-coated sensor-tip surfaces (180 s), followed by a dissociation phase (240 s). To account for potential non-specific interactions, we dipped sensors lacking biotinylated cROS1 into cNEL or cNEL/hNICOL solutions, and to account for dissociation of biotinylated cROS1 from the sensor over the course of the experiment, we dipped the biotinylated cROS1-coated sensor into kinetic buffer lacking analyte followed by subtraction of the signals of both control measurements using the Octet® Analysis Studio software.

## Cryo-EM grids preparation

Quantifoil R1.2/1.3 300-mesh gold holey carbon grids (Quantifoil, Micro Tools GmbH, Germany) were glow-discharged for 60 s with PELCO easiGlow™ Glow Discharge Cleaning System. Cryo-EM grids were prepared with a Vitrobot Mark IV (FEI). For cROS1/cNEL sample, the cryo-EM grid was prepared by applying 3.8 µl of the sample at the concentration of 0.8 mg/ml to glow-discharged grids. Grids were blotted for 3.0 s under 100% humidity at 4 °C before being plunged into nitrogen-cooled liquid ethane using the Mark IV Vitrobot. For cROS1 only sample or cROS1/cNEL/hNICOL sample, the samples were concentrated to 8 mg/ml, and 3 mM fluorinated Fos-Choline-8 (Anatrace)(final concentration) was added to the concentrated sample immediately before preparing the cryo-EM grids. The cryo-EM grids were prepared by applying 3.8 µl of the protein samples to glow-discharged Quantifoil R1.2/1.3 300-mesh gold holey carbon grids. Grids were blotted for 4.0 s under 100% humidity at 4 °C before being plunged into the liquid ethane using the Mark IV Vitrobot.

## Cryo-EM data collection and image processing

EM data acquisition, image processing, and model building, and refinement were performed following previous protocols with some modifications. Micrographs were collected in the counting mode on Titan Krios microscopes (Thermo Fisher Scientific) with K3 Summit direct electron detectors (Gatan). The nominal magnification and pixel size of each data set are summarized in Table 1. Motion-correction and dose-weighting of the micrographs were carried out using the Motioncor2 program (version 1.2)[32]. GCTF 1.06 was used for CTF correction[33]. Template-based particle picking was carried out using the autopick tool in RELION 5.0[34]. Particles were binned with 4 times and cleaned up with multiple rounds of 2D and 3D classification in RELION. The exact procedures are summarized in Supplementary Figs. The initial mode for 3D classification and refinement was generated using the SGD method in RELION. The final refinement was carried out with unbinned particles, and the refined maps were further improved by using Bayesian polishing, blush regularization, and CTF refinement at the final stage. The Fourier Shell Correlation (FSC) 0.143 criterion was used for estimating the resolution of the maps. Local resolution was calculated in RELION.

For the dataset of ligand-free-cROS1, the 3D classification with small angle sampling search revealed three overall conformations (3.7-degree sampling, 20 degrees searching range), with different relative angles between the "head" and "leg" regions of cROS1. After focused refinement using a soft mask around either the "head" or "leg" region, the cryo-EM densities for both the "head" and "leg" regions were improved significantly and were resolved at 2.9 Å and 3.6 Å resolution, respectively.

For the dataset of cROS1/cNEL complex, the initial cryo-EM map of cROS1/cNEL was resolved at 2.7 Å resolution, but only the VWC2 domain of cNEL was resolved. After focused classifications using a soft mask on the top region of the cROS1/cNEL complex, the EGF1, EGF2, VWC1, and VWC2 domains were resolved. The overall resolution for this map is 3.8 Å.

For the dataset of cROS1/cNEL/hNICOL, the part corresponding to the cNEL/hNICOL was poorly resolved in the initial cryo-EM determination. After focused classification with density subtraction[22], the cryo-EM density of 2:1 cNEL/hNICOL complex was improved remarkably. Before the focused refinement of the 2:1 cNEL/hNICOL complex, we centered the density corresponding to the 2:1 cNEL/hNICOL complex to the center of the box and re-extracted all the particles to a smaller box. The focused 3D classification was performed using a smaller angular sampling (3.7 degree) and local search(20 degree). Two different conformations were identified, with one ROS1 molecule bound with each of the cNEL protomers, respectively. The 3D refinement of original particles without density subtraction resulted in cryo-EM maps for two distinct conformations of 1:2:1 cROS/cNEL/hNICOL complex, at the resolution of 4 Å and 4.2 Å, respectively.

## Model building and refinement

Model building was initiated by docking of the AlphaFold predicted models of the individual proteins into the cryo-EM maps. The models were adjusted manually and refined using the ISOLDE plugin in ChimeraX[35,36]. N-glycans were added to the models based on the density using ISOLDE. The coordinates and atomic b-factors of the models were further refined using the real-space refinement module in Phenix[37], with secondary structure and Ramachandram restraints. The validation of the models was carried out using the Molprobity module in Phenix.

## Reporting summary

Further information on research design is available in the Nature Portfolio Reporting Summary linked to this article.

## Data availability

All reagents generated in this study are available with a complete Materials Transfer Agreement. All cryo-EM maps and models reported in this work have been deposited into EMDB/PDB database, under the entry ID: ligand-free-ROS1 head region EMDB EMD-71042; PDB: 9OYZ), ligand-free-ROS1 leg region EMDB: EMD-71047; PDB 9OZ1), ligand-free-ROS1 complete model EMDB EMD-71059; PDB: 9OZI), 1:1 cROS1/cNEL complex EMDB EMD-71049; PDB 9OZ6), 2:1 cNEL/hNICOL complex EMDB EMD-71051; PDB 9OZ8), 1:2:1 cROS1/cNEL/hNICOL holo-complex conformation 1 EMDB EMD-71057; PDB 9OZC), and 1:2:1 cROS1/cNEL/hNICOL holo-complex conformation 1 EMDB EMD-71058; PDB 9OZH). The source data for Figs. 2e, 5c, and Supplementary Fig. S7b, S7c, S8, S9a, S9b, S13b, and S13c are provided in a Source Data file. Source data are provided with this paper.

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

## Acknowledgements

Cryo-EM data were collected at the University of Texas Southwestern Medical Center (UTSW) Cryo-Electron Microscopy Facility, funded in part by the Cancer Prevention and Research Institute of Texas (CPRIT) Core Facility Support Award RP170644. We thank Dr. Stoddard for facility access. We thank Dr. Chad A. Brautigam and Dr. Shih-Chia (Scott) Tso for the assistant of SEC-MALS experiments. We thank the Structural Biology Laboratory at UTSW for equipment use. This work is supported in part by grants from the National Institutes of Health (R35GM130289 to X.Z. and R35GM156386 to X.-c.B.), the Welch foundation (I-1702 to X.Z. and I-1944 to X.-c.B.). X.-c.B. and X.Z. are Virginia Murchison Linthicum Scholars in Medical Research at UTSW.

## Author contributions

X.Z. and X.-C.B. designed and supervised research; W.A. designed and performed biochemical and cell biological experiments. W.A., X.Z., and X.-C.B. performed structural biology experiments, X.Z. and X.-C.B. built and refined structural models. All the authors analyzed data; all the authors contributed to the writing, editing and reviewing of the manuscript.

## Competing interests

The authors declare no competing interests.
