## [Transparent Peer Review file · Nature Communications]

Structural Insights into the Activation of the Chicken ROS1 Receptor by the NEL/NICOL Ligand Complex

Corresponding Author: Dr Xiaochen Bai

Version 0:

Reviewer comments:

Reviewer #1

(Remarks to the Author)

This is an interesting and timely study that advances our understanding of vertebrate (chicken) ROS1 regulation. To date, only *Drosophila* ROS1 (Sevenless) has been structurally and mechanistically characterized, and it employs an entirely different ligand and activation mechanism that is not conserved in vertebrates. Here, the authors use chicken ROS1 (cROS1) and its ligand NEL, with or without the co-ligand NICOL, to determine structures that point towards a novel mode of regulation. The cryo-EM work appears to be of high quality and at resolutions sufficient to define observed interactions. These findings represent an important step toward understanding the molecular basis of vertebrate ROS1 signaling.

However, several major issues need to be addressed before the conclusions can be fully supported.

Major Concerns

1. Supporting biophysical characterization

- o While the structures are well executed, there is no quantitative binding analysis (SPR/BLI) comparing ROS1–NEL and ROS1–NEL/NICOL interactions.
- o No analytical ultracentrifugation (AUC) or SEC-MALS data are provided to define the oligomeric states of the complexes in solution. In particular, SEC-MALS comparisons of ROS1 alone, ROS1–NEL, and ROS1–NEL/NICOL would clarify stoichiometry and complex formation.
- o Even SEC traces for ROS1–NEL/NICOL versus ROS1–NEL are absent.

2. Cell-based activation studies

- o ROS1 phosphorylation is not measured; only downstream ERK phosphorylation is reported. In a transient expression system, direct evidence of ROS1 activation is essential.
- o The fact that point mutants and truncations still show elevated pERK relative to the no-ligand control is puzzling and warrants explanation in the context of proposed mechanism.
- o An inhibitory experiment could strengthen the claims: titrating mutant or truncated NEL/NICOL into wild-type ROS1 activation assays to assess suppression of signal.

3. Limited kinetic and dose–response analysis

- o Activation assays are performed at a single high concentration (100 nM) and a single long time point (25 min). This design precludes understanding the relative kinetics of activation by NEL versus NEL/NICOL.
- o A time course and dose–response analysis could reveal whether NEL/NICOL induces prolonged activation versus NEL inducing transient but robust activation.

4. Unresolved questions about NEL oligomeric state

- o The manuscript does not address whether NEL alone is trimeric, and the observation that NEL alone can activate ROS1 appears inconsistent with an absolute requirement for NICOL and subsequent Laminin G directed polymerization. (If all laminin G domains are engaged in the NEL–ROS1 interface, how could NEL still facilitate higher-order clustering? The structural rationale is unclear.)

5. Structural modeling and interpretation

- o Discussions of laminin G interactions are dense and difficult to follow without clearer figures.
- o The low-resolution density attributed to higher-order assembly via “blob docking” is not convincing.
- o The authors should consider modeling potential dimer-on-dimer arrangements and estimate the resulting kinase domain separations to assess feasibility of activation.
- o Why is the cNEL-ROS1 structure the only structure without the leg modeled? The description states that cNEL binding doesn't cause any changes in cROS1, but this seems to be a potential important change.
- o The density of the second bat-wing of cNEL is less well defined than the protomer that shows clear binding to cROS1. Is it possible that a flexible ROS1 could be bound at the second site? This would be similar to the proposal of lamininG induced clustering (which the authors claim can not be seen directly in the cryoEM because of flexibility) and achieve the same signaling.

6. Ligand species mismatch

- o The use of human NICOL with chicken NEL and chicken ROS1 may be problematic. As seen in the supplied alignments, human NICOL contains an additional cysteine that could significantly alter association and activity. The authors should test chicken NICOL in NEL complex to rule out unintended artifacts.

Minor Comments

- The title should specify “chicken ROS1” or “avian ROS1” to avoid overgeneralizing.
- Replace “apo” with “unliganded” or “ligand-free,” as “apo” is typically used for small-molecule-free enzyme forms.
- Maintain consistent terminology for the unliganded receptor conformation (“extended” vs. “arc-shaped”).
- The phrase “hNICOL promotes cNEL dimerization” may be misleading, as it implies cNEL is otherwise monomeric. Clarify whether cNEL is trimeric, as expected for other NELLs, and whether NICOL prevents trimerization.
- Correct typographical errors: e.g., “cNET/hNICOL” should read “cNEL/hNICOL.”

Overall, the structural data are of high potential interest, but the lack of supporting biochemical and cell-based validation makes it difficult to assess the physiological relevance of the proposed mechanism. Adding quantitative binding studies, oligomeric state analyses, direct ROS1 phosphorylation data, and expanded activation kinetics would substantially strengthen the manuscript.

Reviewer #2

(Remarks to the Author)

The authors present the first full-length extracellular domain structure of a higher vertebrate receptor tyrosine kinase (RTK) ROS1 in its apo form and bound to the ligand NEL, and a ligand/co-ligand complex NEL/NICOL. This is an exciting finding that provides new insight into oligomerization via interaction with this co-liganded complex. Activation of RTKs, which are important regulatory signaling molecules and common therapeutic targets, is a topic of great current interest as we are beginning to understand that RTK signaling often has several layers of complexity, and this paper provides some of the first direct structural evidence for clustering mediated by co-ligands, which I think will be of great interest to your readership and something that is widely discussed and cited going forward.

The authors provide a careful analysis of the three structures and follow up key findings with functional assays. It's also interesting that sugars are clear mediators of some domain interactions in apo ROS1, highlighting the importance of determining these structures with glycans included, as the authors did here. This is worth mentioning, because historically many RTK structures are deglycosylated, a practice which has clearly hidden important features of structural organization. I've included a few comments below that I thought were important to make:

The authors comment about missing domains from cNEL in the cROS1/cNEL complex, but not on the missing “leg” of the ECD domain of cROS1. Was this also likely due to conformational flexibility, as informed by analysis of 2D classes, or another possible reason? They indicate in the discussion that ROS1 does not undergo significant conformational changes upon NEL binding, but it's unclear that this statement can be made without the entire ECD. Increased head to leg flexibility, for example, could be significant and functionally important. While I don't think this structure needs to have the leg domain (and it was likely challenging), it would be nice to see some comments and thoughts informed by their experimental data on what may be happening here, especially as leg domain movement would likely be important for bringing intracellular domains in close proximity in oligomers. Of note, more of this is visualized in the co-liganded complex, which I think suggests that the ligands may have different effects on full-length ROS1.

While the authors provided pdb validation reports that for the most part look excellent (90ZH and 90ZC appear to have some validation issues, but this is not uncommon for lower resolution structures), they did not provide any raw data (pdb and generated maps)? It would be best for reviewers to see this data before being sure about their review, as it would be helpful to determine what may have been the source of issues in the validation report (was it truly a function of the resolution or other potential issues). Obviously, due to size we cannot generally share all EM raw data, but the final maps and pdbs are easy to share.

In the same note, I assume the residues substituted for functional analysis (i.e., F318A, E541A, F527A and Y553A) were surface exposed and unlikely to impact the tightly bent structure of the ROS1 “head”, but I cannot determine this without the pdbs. If they are not surface exposed or suspected effects on packing could occur, it would suggest impacts to activation could be due to local instability rather than disrupted interactions with ligand. Figure 2D does show these residues, but it's impossible to use this image for this type of analysis. For example, E541 looks like it has a potential ionic interaction with R368 also of ROS1 such that mutating this to alanine may cause local instability. As such, a clear interpretation that these substitutions impacted interaction with ligand and not folding would be difficult to make. However, given that they mutated several sites independently, it seems likely that ligand interaction is being impacted, making this a less critical point.

The color choices for the NEL/NICOL complex (especially in Figure 4) make it quite hard to distinguish the two, as there could be more contrast.

The authors argue that liganded ROS1 likely forms higher order oligomers, or arrays, which has been suggested in the activation of other RTKs and is a current hot topic of interest in the field (making this paper highly relevant), but they do not model this in their working model (Figure 6). I think this is a big oversight, as someone reading this figure would just think this represents a model for simple ligand-induced dimerization. Figure 6 would also be stronger as a standalone figure if the complex components were labeled or if there was some sort of legend.

Lastly, I wonder if the authors tried modeling the dimer using the additional LamG density as a start and then tried to create templates from that to see if it was found in the dataset? The additional density does seem to suggest dimers were present in the sample, but at a lower sampling rate. Given that the monomeric complex structure seems pretty rigid, this may be worth trying if they haven't already. It may provide a bit more information about the receptor:ligand/co-ligand 'dimer'.

Reviewer #3

(Remarks to the Author)

Overall Assessment

This manuscript reports the cryo-EM structure of the NEL–NICOL–ROS1 complex, providing high-resolution structural insight into ROS1 activation by its recently identified endogenous ligands, NELL2 and NICOL. While ROS1 has long been recognized in oncology as a receptor tyrosine kinase activated by oncogenic fusion proteins, the physiological ligands and activation mechanisms have remained unclear until recent work identified the NELL2/NICOL complex as a key mediator of testis-to-epididymis signaling and male fertility. From a structural biology perspective, no detailed architecture of this ternary complex has been described, making the present work timely and potentially significant.

The study's key strength lies in the novelty of the structural findings, which extend our understanding of ROS1 signaling beyond the oncogenic context and provide a foundation for exploring distinctive modes of RTK activation. The cryo-EM data reveal a ligand–co-ligand–receptor arrangement that offers a structural framework likely to inspire new hypotheses about receptor clustering and signal initiation. The topic will interest structural biologists, reproductive biologists, and the broader RTK signaling community.

Several aspects would benefit from further attention to maximize impact and clarity. The methodological description—particularly sample preparation and the selection/heterogeneity of conformational states for cryo-EM analysis—is not yet sufficiently detailed to fully assess reproducibility or the basis of structural interpretation. The linkage to prior biochemical/structural literature could be sharpened; the novelty would be even clearer if the authors positioned their results explicitly relative to earlier models and explained methodological differences that may account for discrepancies. In addition, some mechanistic claims, especially regarding ROS1 activation, would benefit from being phrased with greater caution to strengthen their persuasiveness.

In summary, this is a potentially high-impact and original study. By adding methodological detail, more explicitly situating the findings within prior literature, and tempering mechanistic proposals where direct evidence is not yet available, the authors can enhance rigor, reproducibility, and interpretability. Addressing the points below should help ensure the work's significance is fully appreciated by a broad readership.

Note on referencing: As the submitted PDF lacks line numbering, specific locations are indicated using figure/panel numbers, section headings, and short quoted phrases rather than page/line numbers.

Major Comments

NICOL-free cNEL

1. According to Extended Data Fig. 1, cROS and cNEL were purified by Ni-affinity chromatography followed by gel filtration on a Superose 6 Increase 10/300 GL column, with peak fractions collected and concentrated. Please indicate whether the gel-filtration step may enrich particular oligomeric states, as this could influence interpretation of subsequent structural analyses.

2. For recombinant NICOL-free cNEL, please clarify whether it exists primarily as a monomer or dimer. Under non-reducing SDS-PAGE it appears not to be monomeric, and Extended Data Fig. 1 suggests a major disulfide-linked dimer plus some monomer. Stating this explicitly in the Results would aid interpretation of complex-formation data.

3. While the data indicate that NICOL-free cNEL (~240 kDa) is not a monomer, SDS-PAGE alone may not be sufficient to conclude it is a dimer. If feasible, please consider confirming the oligomeric state by mass spectrometry or an equivalent technique to strengthen the conclusion.

4. If NICOL-free cNEL does form a dimer, please clarify whether the LamG–VWC4 interaction occurs between protomers. Indicating whether this is clearly observed, unclear, or unresolved would help readers assess the mechanism.

cNEL–cROS interaction

5. Extended Data Fig. 1 shows that NICOL-free cNEL used for complex formation with ROS1 was disulfide-linked dimeric rather than monomeric. Explicitly stating this in the Results would contextualize complex formation.

6. For the cROS–cNEL complex, purified proteins were mixed, separated by gel filtration, and peak fractions were analyzed (Extended Data Fig. 1). Were the SDS-PAGE analyses performed under reducing conditions? If so, what is the band just below cROS—could it be dimeric cNEL or a partially reduced form? Clarifying the analysis conditions and band assignments will aid interpretation.

7. In the cryo-EM analysis, only the cNEL molecules adjacent to cROS are visible. It remains possible that a cNEL dimer could bind two cROS molecules, particularly if cNEL is more flexible in the absence of NICOL. A brief discussion of this alternative would help frame the observations.

8. The gel-filtration profile for cROS–cNEL shows an additional early-eluting peak around column volume ~8. Could this correspond to a cNEL dimer bound to two cROS molecules? Providing an interpretation would be informative for readers considering alternative stoichiometries.

cNEL–hNICOL interaction

9. Please discuss, even briefly, structural differences between previously reported NELL2/NICOL complexes and the complex proposed here. Indicating whether differences are methodological, biologically relevant, or both would help contextualize the novelty.

10. Were there prior predictions or preliminary data suggesting that NELL2 and NICOL might form a covalent complex? If so, summarizing them would provide helpful context.

11. For the statement, “Consistent with previous findings, our cell-based assays demonstrated that the cNEL/hNICOL complex activates ROS1 much more robustly than cNEL alone,” please verify that the supporting citations are accurate and that the prior findings are represented precisely.

cNEL–hNICOL–cROS interaction

12. Prior studies reported NELL2 as a trimer, with no evidence of a covalent NELL2–NICOL complex. The finding here of a covalent 2:1 NELL2–NICOL complex is highly novel. Explicitly citing the prior work and highlighting the differences would underscore originality.

13. The observation that the NELL2–NICOL complex has two ROS1-binding sites, yet no complex with both sites occupied was observed, may relate to sample preparation—specifically, collecting fractions containing only singly bound complexes (Extended Data Fig. 1). Acknowledging this possibility would temper conclusions about stoichiometry.

14. The manuscript states that cNEL is “flexible and monomeric” in the cROS–cNEL complex but “rigid and dimeric” in the presence of hNICOL. Given that the cNEL used for cROS–cNEL complex formation was already dimeric, please reconcile these statements to ensure internal consistency.

15. The steric-clash model preventing simultaneous binding of two ROS1 molecules to a cNEL dimer is interesting. Please specify the extent/nature of the predicted overlap to help readers assess the basis of this conclusion.

ROS1 activation mechanism

16. The proposed activation model involving higher-order multimerization via VWC4–LamG interactions is intriguing but remains speculative based on the current data. Consider moving this to the Discussion, with appropriate caveats, and note explicitly that direct structural evidence is not yet available.

17. Please clarify how cNEL and cNEL–hNICOL mutants used in BN-PAGE and cell-based assays were prepared. In particular, indicate whether gel filtration was used to isolate fractions of defined oligomeric state, which is essential for evaluating the activation mechanism.

18. The finding that cNEL– Δ LamG–hNICOL fails to form multimers in BN-PAGE and cannot activate ROS1 despite binding it is mechanistically important. Emphasizing this in the Results and linking it clearly to the proposed model would be helpful.

19. In the section where the authors state that “the cNEL– Δ CC mutant predominantly existed in a monomeric state,” yet BN-PAGE shows multimers: if only monomeric fractions were used for the cell-based assay, this could explain the lack of activation. Clarification would aid interpretation.

20. Since NICOL-free cNEL can form dimers, could it also form higher-order assemblies via VWC4–LamG interactions to activate ROS1? If so, what is the role of hNICOL in this model, particularly with respect to VWC5–NICOL/CC interactions? If unresolved, stating this explicitly would be useful.

21. The mechanism proposes that multiple ROS1 molecules bind a single ligand assembly and approach one another sufficiently for intermolecular phosphorylation. As this arrangement is not directly visualized, acknowledging this limitation would strengthen the Discussion.

22. The manuscript contrasts a “classical RTK” model with a distinct ROS1 activation mechanism. This perspective is potentially correct, but conclusions should be tempered to reflect remaining uncertainties and avoid overstatement.

23. The statement that “ROS1 is, to date, the only known RTK that depends on a co-ligand, NICOL, for activation” is not precise, as FGFR activation by FGF requires heparin/heparan sulfate, which function as co-ligands. Please revise accordingly.

24. The claim that NICOL “stabilizes cNEL into a rigid conformation optimal for ROS1 activation” could be reconsidered. The data may instead support higher-order oligomerization via LamG–VWC4 interactions as the critical determinant. Clarifying this distinction would avoid overinterpretation.

Minor Comments

25. For cROS1-H3C-Tsi-His8, hNICOL-H3C-Tsi-His8, cNEL-H3C-Tsi-His8, cNEL– Δ LamG, and cNEL– Δ CC, please provide the amino-acid sequences of linkers/tags appended to the C-terminus in Methods to facilitate reproducibility.

26. In the section discussing the *Drosophila* ROS1 homolog Sevenless (Sev) and its ligand Boss, the content is interesting but not directly tied to the manuscript’s core novelty. In addition, the statement “we did not observe cROS1 dimerization under acidic conditions” lacks supporting data. Consider shortening this section or moving it to Supplementary Information.

27. In Fig. 2a and Fig. 3e, the numbering/labeling of VWC domains appears incorrect. Please correct for accuracy and clarity.

Version 1:

Reviewer comments:

Reviewer #1

(Remarks to the Author)

Review of: “Structural Basis for the Activation of the Chicken ROS1 Receptor by the NEL/NICOL Ligand Complex” by An et al.

The revised draft adds data and at the same time has toned down some speculation that was without supporting data. Although, I would suggest additional toning down on speculation, and reexamining new data. Below are suggestions that could further clarify points for the readers and make for a more polished manuscript. In particular, Fig 6 could be made more impactful to show the thoughts of the authors in their new Discussion section, where they speculate on the impact of NICOL on activation.

Important control (critical)

Fig 2E, 4c and cell activation throughout (Supp 7, Supp 8, Supp 9, Supp 13). There are no controls for ligand alone (cNEL or cNEL/NICOL) tested to parental cells (HEK or H1299) without cROS1 expression. Unless I missed this? Since NEL is known to interact with another receptor ROBO3 (and potentially additional receptors), downstream ERK could come from many sources. This is a necessary control.

Logic missing regarding the comparisons made between cNEL vs cNEL/NICOL induced activation/mechanism:

What is the proposed mechanism for how NICOL changes signaling of cNEL? Reader is left confused about how NICOL affects function. It is ok to state that it remains yet unknown, and that imparted rigidity may play a factor, and show this as hypothesis in Fig 6. Compared to without NICOL.

From the data: both ligands (NEL dimer and NEL/NICOL heterotrimer) bind equivalently to ROS1, both form higher order oligomers, both have a foundational dimer of cNEL in the complex. Yet robust signaling requires NICOL. The authors essentially present structures in the manuscript of unliganded ROS1 (Fig 1), ligand bound but inhibited (*partly) ROS1 (NEL only) (Fig 2), and ligand bound active ROS1 (NEL+NICOL) (Fig 3). From the visual presentation of figures the logic rather seems: 1) with no ligand ROS1 has a slightly flexible leg region, 2) with cNEL there is more flexibility, whereas 3) with cNEL plus NICOL, ROS1 has a rigidly linked leg region. This, coupled to cNEL's oligomerization, seems most supported by the structural analysis as a link to function. Authors should mention some of this directly, as the reader will be drawn to these clear structural differences.

Fig 6. Too much attention paid to the speculative (3 different) ways of oligomerization, and this only focuses on with NICOL present. How does NICOL effect compared to what happens without NICOL? This figure should reflect the new Discussion. Fig6 should show what is supported by data: how NEL interacts with ROS1, how this is NEL dimer (with or without NICOL) bound to 1 ROS1, and that some yet to be determined mechanism (not these 3 speculative but similar ways) leads to higher order clustering. With a "?" showing this resolution still not there. Real question throughout manuscript is how NICOL affects activity. This should be compared/contrasted to the minus NICOL cluster model.

NEL dimer vs trimer

Authors state that cNEL is a dimer, with or without NICOL present. This is different than mammalian NELL2 or (NELL1) where there is clear evidence that shows NELLs are trimeric: (Nature Communications 2020, PMID: 32198364). Thus, a dimer is unexpected and not consistent with prior reports. This is interesting, as avian NEL is likely different than mammalian NELLs. Authors should point out this interesting finding.

Issues with the SECMAALLs in Fig 5.

Some of the masses predicted exceed a megadalton which is not consistent with the elution volume. These peaks are clearly not at the void volume of column (~15min), yet predicted masses for oligomers far exceed size limitation separated by column. Please explain.

Also, why did authors only show UV for cNEL/hNICOL? The UV signal shows almost none of the protein to be oligomeric. What about UV traces for others? cNEL alone, seems more significant fraction is oligomer (by LS). For cROS1 what is the shown trace? Is it LS or UV, because LS labeled purple, but curve is yellow.

When one focuses on the linear (non-oligomer) mass predicted of each: it seems cNEL ~300 kDa, cNEL/hNICOL ~300 kDa, and cROS1 ~300 kDa. Is this consistent with authors prediction of dimer cNEL, dimer cNEL+ monomer NICOL, and monomer cROS1? If one then compares these masses to the SECMAALLs of complexes (cROS1/cNEL with mass ~900kDa, and cROS1/cNEL/hNICOL with mass ~600kDa), this indicates that without NICOL, 2 ROS1's can actually bind to one cNEL (300+300+300=900). Whereas, with NICOL, only 1 ROS1 can bind (300+300=600). The authors should examine this data for their interpretation. This could fit their structural findings that seem to show increased rigidity with NICOL, and how this might limit a second ROS1 binding event.

Minor:

The title: "Structural basis for the Activation of the Chicken ROS1 Receptor by the NEL/NICOL Ligand Complex" implies that the structural basis for activation has been elucidated. However, the certainty described here is rather for how cNEL engages cROS1. Not for activation per say. The exact structural mode of driving activation through clustering remains to be defined.

Should Supplemental Figs be called Extended Data?

The statements that ROS1 adopts an "extended conformation" is a bit misleading. The N-terminus is in the bent-over conformation as described for Sevenless. "Extended" seems to imply an opening up of this "bent-over" state. The C-terminus

'leg' is an extension from this. But one that is not rigid in the absence of ligand. So better described as similar to Sev, N-terminus is folded back, and C-terminal half flexible and dynamic.

Throughout, the N-term domain referred to as HD for helical domain, but this region has been previously referred to as a CATCH domain for (cysteine-rich ALKAL-type coupled Helices) for the drosophila Sevenless receptor.

Line 23: " Structural analyses reveal that the 2:1 NEL/NICOL complex further oligomerizes through LamG–VWC4 domain interactions, facilitating the clustering of multiple ROS1 for its activation."

Does the authors' "structural analysis" actually "reveal" this? What structural work unambiguously revealed this interaction? Or rather is biochemical experiments suggestive of this?

Line 77: Despite the present (*presence) of two cNEL protomers in this complex,

Line 106: "A similar D-shaped fold has been observed in the structure of Sevenless (Sev), the Drosophila ortholog of ROS1, indicating the structural conservation of ROS1 across species." No citation, or structural comparison made here?

Line 246: "SEC-MALS and blue native PAGE (BN-PAGE) results showed ... cNEL/hNICOL and cROS1/cNEL/hNICOL complex can further assemble into higher-order oligomers (Fig. 5a,b)."

But data also shows cNEL and cNEL-ROS1 (without NICOL) can form same higher order oligomers. Therefore, stating this is in isolation is misleading one to believe that cNEL requires NICOL for additional oligomerization.

Reviewer #2

(Remarks to the Author)

As noted previously, The authors present the first full-length extracellular domain structure of a higher vertebrate receptor tyrosine kinase (RTK) ROS1 in its apo form and bound to the ligand NEL, and a ligand/co-ligand complex NEL/NICOL. This is an exciting finding that provides new insight into oligomerization via interaction with this co-liganded complex. Activation of RTKs, which are important regulatory signaling molecules and common therapeutic targets, is a topic of great current interest as we are beginning to understand that RTK signaling often has several layers of complexity, and this paper provides some of the first direct structural evidence for clustering mediated by co-ligands, which I think will be of great interest to your readership and something that is widely discussed and cited going forward. Overall, I think the authors have responded adequately to criticisms and questions offered by myself and other reviewers. They provided model and map data that allowed me to answer several of the questions I posed as well. I think the changes to Figure 6 have enhanced the presentations of their conclusions/hypotheses formed by their work. One last comment to the authors: I appreciate their efforts to visualize a receptor:ligand/co-ligand dimer, and just offer one more suggestion to try running ab-initio multiple times where you vary the number of classes (up to 10) to see if they might be able to pull out the dimer just in case this has not been tried. This is not a suggestion within the scope of this publication as they have likely tried this and I think the manuscript already provides significant advances that will be of great interest to numerous readers.

Version 2:

Reviewer comments:

Reviewer #1

(Remarks to the Author)

Reviewer #1 (Remarks to the Author):

This is an interesting and timely study that advances our understanding of vertebrate (chicken) ROS1 regulation. To date, only *Drosophila* ROS1 (Sevenless) has been structurally and mechanistically characterized, and it employs an entirely different ligand and activation mechanism that is not conserved in vertebrates. Here, the authors use chicken ROS1 (cROS1) and its ligand NEL, with or without the co-ligand NICOL, to determine structures that point towards a novel mode of regulation. The cryo-EM work appears to be of high quality and at resolutions sufficient to define observed interactions. These findings represent an important step toward understanding the molecular basis of vertebrate ROS1 signaling.

We thank the reviewer for the positive assessment of our manuscript.

However, several major issues need to be addressed before the conclusions can be fully supported.

Major Concerns

1. Supporting biophysical characterization

o While the structures are well executed, there is no quantitative binding analysis (SPR/BLI) comparing ROS1–NEL and ROS1–NEL/NICOL interactions.

Response: Good point. In response to the reviewer's comment, we analyzed the binding affinity using BLI. The results showed that ROS1–NEL and ROS1–NEL/NICOL have nearly identical binding affinities, with the K_d values in the single-digit nanomolar range. These new results have been added to the revised manuscript (new Supplementary Figure 3).

o No analytical ultracentrifugation (AUC) or SEC-MALS data are provided to define the oligomeric states of the complexes in solution. In particular, SEC-MALS comparisons of ROS1 alone, ROS1–NEL, and ROS1–NEL/NICOL would clarify stoichiometry and complex formation.

Response: Thanks for the good suggestions. We have performed SEC-MALS characterization on cNEL, cNEL/hNICOL, cROS1, cROS1/cNEL, and cROS1/cNEL/hNICOL sample. The results showed that cROS1 alone is a monomer, but forming large high-order oligomer when bound to either cNEL or cNEL/hNICOL complex. cNEL or cNEL/hNICOL can also self-assemble into a higher-order complex. All these results are consistent with our blue native PAGE results, supporting our proposed activation model. These new results have been included in the new Figure. 5a.

o Even SEC traces for ROS1–NEL/NICOL versus ROS1–NEL are absent.

Response: Point accepted. In the revised manuscript, we have included the SEC result of the ROS1–NEL/NICOL complexes and compared its SEC profile with that of ROS1–NEL (new Supplementary Figure 6).

2. Cell-based activation studies

o ROS1 phosphorylation is not measured; only downstream ERK phosphorylation is reported. In a transient expression system, direct evidence of ROS1 activation is essential.

Response: Thank you to the reviewer for raising this important point. We initially examined cROS1 phosphorylation in 293T cells but did not observe any difference between the cNEL/hNICOL-treated and control groups. We then tested ROS1 phosphorylation in H1299 cells, where the results showed that cNEL/hNICOL moderately stimulates cROS1 and ERK phosphorylation (new Supplementary Figure 8). Although the increase in ROS1 phosphorylation was modest, the results were reproducible and statistically significant. Notably, in the original study that identified NELL2 as a ROS1 ligand (PMID: 32499443), the authors reported only ERK phosphorylation rather than ROS1 phosphorylation, suggesting that direct detection of ROS1 phosphorylation upon NELL2 stimulation is technically challenging. We speculate that, although ROS1 autophosphorylation is relatively weak, signaling strength may be amplified through the downstream pathway. Given the weak autophosphorylation of ROS1, we chose to evaluate all mutants based on their effects on ERK phosphorylation in 293T cells.

o The fact that point mutants and truncations still show elevated pERK relative to the no-ligand control is puzzling and warrants explanation in the context of proposed mechanism.

Response: Thank you to the reviewer for raising this important point. Our new BLI results showed that cNEL binds to cROS1 with extremely high affinity ($K_d \approx 2$ nM). We suspect that the point mutation alone may not be sufficient to completely disrupt the interaction between cROS1 and cNEL. Furthermore, our native gel results indicated that the cNEL- Δ CC/hNICOL and cNEL- Δ LamG/hNICOL truncations still displayed higher-order oligomerization, albeit at a lower level than the wild type complex (Fig. 5b). These results are consistent with the weakened but not abolished pERK elicited by the point mutants and truncations.

o An inhibitory experiment could strengthen the claims: titrating mutant or truncated NEL/NICOL into wild-type ROS1 activation assays to assess suppression of signal.

Response: Thank you to the reviewer for this excellent suggestion. We titrated a NEL mutant lacking both the Coiled-Coil and LamG domains (cNEL- Δ CC- Δ LamG) into wild-type cNEL cROS1 activation assays. Our results showed that cNEL- Δ CC- Δ LamG inhibited WT cNEL-induced cROS1 activation in a dose-dependent manner. This finding further supports that higher-order oligomerization of the cNEL/hNICOL complex, in addition to receptor binding, is required for cROS1 activation. These new results have been included in the revised manuscript (new Supplementary Figure 13c).

3. Limited kinetic and dose–response analysis

o Activation assays are performed at a single high concentration (100 nM) and a single long time point (25 min). This design precludes understanding the relative kinetics of activation by NEL versus NEL/NICOL.

o A time course and dose–response analysis could reveal whether NEL/NICOL induces prolonged activation versus NEL inducing transient but robust activation.

Response: Thank the reviewer for this excellent suggestion. Following the suggestion, we performed cell-based assays using a range of ligand concentrations. The results showed that the activation of ROS1 reaches plateau at around 100 nM concentration of cNEL/hNICOL. In contrast, cNEL cannot robustly induce ROS1 activation even at concentrations up to 2000 nM. These results further demonstrated the critical role of NICOL in NEL induced ROS1 activation (new Supplementary Figure 9a). Furthermore, our time course analysis indicated that the maximal activation of ROS1 by either cNEL/hNICOL complex or cNEL alone was reached at ~30 min (new Supplementary Figure 9b), although the efficacy in ROS1 activation by cNEL alone was much weaker. These data further suggest that the NEL alone is not optimal for ROS1 activation.

4. Unresolved questions about NEL oligomeric state

o The manuscript does not address whether NEL alone is trimeric, and the observation that NEL alone can activate ROS1 appears inconsistent with an absolute requirement for NICOL and subsequent Laminin G directed polymerization. (If all laminin G domains are engaged in the NEL–ROS1 interface, how could NEL still facilitate higher-order clustering? The structural rationale is unclear.)

Response: Good point. Our SDS-PAGE results show that cNEL alone primarily exists as a disulfide-linked dimer and can further oligomerize, capable of activating ROS weakly. We speculate that NICOL may facilitate the formation of higher-order receptor–ligand assemblies that position multiple ROS1 molecules in a more favorable configuration for more potent activation. However, this remains speculative, as such higher-order complexes were not directly observed in our cryo-EM analysis. In addition, our biochemical and cryo-EM data indicate that cNEL exhibits markedly improved biochemical stability and reduced aggregation in the presence of NICOL, suggesting that NICOL may enhance the biological activity of NEL by stabilizing its structure and improving its solubility. Interestingly, NEL and NICOL display highly similar expression patterns across tissues. For example, they are both highly expressed in the brain and male reproductive tissues. This raises the intriguing possibility that NEL is always associated with NICOL in vivo and that NICOL functions as a constitutive structural partner of NEL. We have briefly discussed these points and their limitations in the revised manuscript.

We apologize for the confusion regarding the NEL-ROS1 interface. The VWC2 domain of cNEL, not the LamG, is required for cROS1 binding. Based on our truncation results, we propose that the LamG domain is critical for mediating higher-order clustering.

5. Structural modeling and interpretation

o Discussions of laminin G interactions are dense and difficult to follow without clearer figures.
o The low-resolution density attributed to higher-order assembly via “blob docking” is not convincing.
o The authors should consider modeling potential dimer-on-dimer arrangements and estimate the resulting kinase domain separations to assess feasibility of activation.

Response: Thank you to the reviewer for raising this critical point. We agree with the reviewer that the quality of our higher-order ROS1/NEL/NICOL complex structure is insufficient for precise modeling. In the revised manuscript, we have moved Fig. 5a and 5b to the supplementary materials and substantially

reduced the discussion regarding how the second 1:2:1 ROS1/NEL/NICOL complex engages the first through LamG-mediated interactions.

In the revised manuscript, we have rewritten the section of “Proposed activation mechanism”. We now propose an activation model for ROS1 based on our native gel and SEC-MALS results, which demonstrate that cNEL/hNICOL and cROS1/cNEL/hNICOL can further assemble into higher-order complexes. We briefly note that, based on the size and shape of the additional density, the higher-order assembly may be mediated by LamG interactions, a hypothesis partially supported by our LamG truncation data. Finally, we have added to the discussion in the revised manuscript that further structural studies will be required to elucidate the molecular details underlying the assembly of higher-order ROS1/NEL/NICOL complexes.

o Why is the cNEL-ROS1 structure the only structure without the leg modeled? The description states that cNEL binding doesn't cause any changes in cROS1, but this seems to be a potential important change.

Response: We thank the reviewer for raising this issue. The “leg” region was not resolved in the cryo-EM structure of the cROS1-cNEL complex. We do not believe that cNEL binding increases conformational flexibility between the head and leg regions of cROS1, because: (1) the cNEL binding site on ROS1 is far from the joint between the head and leg regions, and (2) the leg region of ROS1 was resolved in the cryo-EM structure of the cROS1/cNEL/hNICOL complex, showing no structural changes compared to ROS1 alone.

We propose an alternative explanation for the poorly resolved ROS1 leg in the cROS1/cNEL complex. Both our BN-PAGE and new SEC-MALS results (Fig. 5) indicate that cNEL alone exhibits poor biochemical behavior, being prone to aggregation. We also observed that cNEL alone tends to precipitate when concentrated to high concentrations. Consequently, the cROS1/cNEL complex may be suboptimal for rapid freezing and more likely to contact the air–water interface on the cryo-EM grid, making it susceptible to damage by the hydrophobic air-water interface. Given the elongated shape of cROS1, this damage could increase dynamics between the head and leg regions, leaving only the rigid core of cROS1 resolved in the cryo-EM map.

o The density of the second bat-wing of cNEL is less well defined than the protomer that shows clear binding to cROS1. Is it possible that a flexible ROS1 could be bound at the second site? This would be similar to the proposal of lamininG induced clustering (which the authors claim can not be seen directly in the cryoEM because of flexibility) and achieve the same signaling.

Response: We thank the reviewer for raising this interesting question. In our cryo-EM analysis, we resolved two distinct 1:2:1 cROS1/cNEL/hNICOL complexes, in which a single ROS1 molecule alternately binds to either cNEL protomer 1 or 2. Our structures further indicate that ROS1 binding stabilizes the conformation of the bound NEL protomer: (1) in conformation 1, protomer 1 is better resolved than protomer 2; (2) in conformation 2, protomer 2 is better resolved than protomer 1.

If both NEL protomers were simultaneously bound to ROS1, their cryo-EM densities would be expected to be equally well resolved; however, such particles were not observed in our dataset. We propose that this binding stoichiometry is unlikely due to steric clashes between the two ROS1 molecules when bound to the same 2:1 NEL/NICOL complex. Nevertheless, we cannot rule out the possibility that a 2:1 NEL/NICOL complex could recruit two ROS1 molecules following significant conformational changes in either NEL/NICOL or ROS1. This possibility has been discussed in the revised manuscript.

6. Ligand species mismatch

o The use of human NICOL with chicken NEL and chicken ROS1 may be problematic. As seen in the supplied alignments, human NICOL contains an additional cysteine that could significantly alter association and activity. The authors should test chicken NICOL in NEL complex to rule out unintended artifacts.

Response: We thank the reviewer for raising this important point. To address it, we co-expressed and purified chicken NEL (cNEL) and chicken NICOL (cNICOL). Cell-based activity assays showed that the cNEL/cNICOL co-ligand complex activated ROS1 to a level very similar to that of the cNEL/hNICOL complex (new Supplementary Figure 7). These results suggest that it is unlikely that ligand species mismatch could cause any structural artifacts. As shown in the figure below, the extra cysteine residue in human NICOL (Cys69) is located in the middle part of the helix and does not make any interaction with either NICOL itself or cNEL. This residue is not conserved in other species such as rat, mouse and chicken, suggesting that it is not a functionally important residue.

Minor Comments

- The title should specify “chicken ROS1” or “avian ROS1” to avoid overgeneralizing.

Response: Point accepted.

- Replace “apo” with “unliganded” or “ligand-free,” as “apo” is typically used for small-molecule–free enzyme forms.

Response: Corrected.

- Maintain consistent terminology for the unliganded receptor conformation (“extended” vs. “arc-shaped”).

Response: Point accepted.

- The phrase “hNICOL promotes cNEL dimerization” may be misleading, as it implies cNEL is otherwise monomeric. Clarify whether cNEL is trimeric, as expected for other NELLs, and whether NICOL prevents trimerization.

Response: Good point. Our SDS-PAGE results showed that cNEL alone exists primarily as a dimer. We also observe small amount of trimers. This is consistent with previous results of NELL1, showing the coexistence of both dimer and trimer (PMID: 24563467, 21814724). We have clarified this point in the revised manuscript.

- Correct typographical errors: e.g., “cNET/hNICOL” should read “cNEL/hNICOL.”

Response: Corrected.

Overall, the structural data are of high potential interest, but the lack of supporting biochemical and cell-based validation makes it difficult to assess the physiological relevance of the proposed mechanism. Adding quantitative binding studies, oligomeric state analyses, direct ROS1 phosphorylation data, and expanded activation kinetics would substantially strengthen the manuscript.

Response: We thank the reviewer for the constructive comments. We hope that the reviewer agree that the additional experiments included the revised manuscript as suggested have substantially strengthened the manuscript.

Reviewer #2 (Remarks to the Author):

The authors present the first full-length extracellular domain structure of a higher vertebrate receptor tyrosine kinase (RTK) ROS1 in its apo form and bound to the ligand NEL, and a ligand/co-ligand complex NEL/NICOL. This is an exciting finding that provides new insight into oligomerization via interaction with this co-liganded complex. Activation of RTKs, which are important regulatory signaling molecules and

common therapeutic targets, is a topic of great current interest as we are beginning to understand that RTK signaling often has several layers of complexity, and this paper provides some of the first direct structural evidence for clustering mediated by co-ligands, which I think will be of great interest to your readership and something that is widely discussed and cited going forward.

The authors provide a careful analysis of the three structures and follow up key findings with functional assays. It's also interesting that sugars are clear mediators of some domain interactions in apo ROS1, highlighting the importance of determining these structures with glycans included, as the authors did here. This is worth mentioning, because historically many RTK structures are deglycosylated, a practice which has clearly hidden important features of structural organization. I've included a few comments below that I thought were important to make:

Response: We thank the reviewer for the positive assessment of our manuscript.

The authors comment about missing domains from cNEL in the cROS1/cNEL complex, but not on the missing “leg” of the ECD domain of cROS1. Was this also likely due to conformational flexibility, as informed by analysis of 2D classes, or another possible reason? They indicate in the discussion that ROS1 does not undergo significant conformational changes upon NEL binding, but it's unclear that this statement can be made without the entire ECD. Increased head to leg flexibility, for example, could be significant and functionally important. While I don't think this structure needs to have the leg domain (and it was likely challenging), it would be nice to see some comments and thoughts informed by their experimental data on what may be happening here, especially as leg domain movement would likely be important for bringing intracellular domains in close proximity in oligomers. Of note, more of this is visualized in the co-liganded complex, which I think suggests that the ligands may have different effects on full-length ROS1.

Response: We thank the reviewer for raising this important question. Reviewer 1 also raised a similar concern. It is important to note that the cNEL alone sample exhibits poorer biochemical properties compared to the cNEL/hNICOL complex, tending to aggregate and precipitate, especially at high concentrations. These observations suggest that the cROS1/cNEL complex is less stable in the absence of NICOL, which may be exacerbated by damage at the hydrophobic air–water interfaces on cryo-EM grids. We believe that these factors account for the increased flexibility between the head and leg regions of cROS1 in the cROS1/cNEL complex, although it is difficult to formally rule out that NELL2 could contribute to it through an allosteric mechanism. However, as the reviewer noted, the leg region of ROS1 is well resolved in the cryo-EM structure of the cROS1/cNEL/hNICOL complex, showing no structural changes compared to ROS1 alone. This observation argues against such an allosteric mechanism.

While the authors provided pdb validation reports that for the most part look excellent (90ZH and 90ZC appear to have some validation issues, but this is not uncommon for lower resolution structures), they did not provide any raw data (pdb and generated maps)? It would be best for reviewers to see this data before being sure about their review, as it would be helpful to determine what may have been the source of issues in the validation report (was it truly a function of the

resolution or other potential issues). Obviously, due to size we cannot generally share all EM raw data, but the final maps and pdbs are easy to share.

Response: Agreed. We have provided all cryo-EM maps and PDB models reported in this work to the reviewers for evaluation.

In the same note, I assume the residues substituted for functional analysis (i.e., F318A, E541A, F527A and Y553A) were surface exposed and unlikely to impact the tightly bent structure of the ROS1 “head”, but I cannot determine this without the pdbs. If they are not surface exposed or suspected effects on packing could occur, it would suggest impacts to activation could be due to local instability rather than disrupted interactions with ligand. Figure 2D does show these residues, but it’s impossible to use this image for this type of analysis. For example, E541 looks like it has a potential ionic interaction with R368 also of ROS1 such that mutating this to alanine may cause local instability. As such, a clear interpretation that these substitutions impacted interaction with ligand and not folding would be difficult to make. However, given that they mutated several sites independently, it seems likely that ligand interaction is being impacted, making this a less critical point.

Response: Good point. All residues selected for mutagenesis are located on the surface of cROS1, unlikely involved in cROS1 folding. Moreover, as the reviewer noted, all point mutations caused similar defects in cROS1 activation, supporting the idea that these mutations disrupt cROS1–cNEL binding rather than affecting the structural integrity of ROS1. Finally, all corresponding PDBs have been provided to the reviewers for further evaluation.

The color choices for the NEL/NICOL complex (especially in Figure 4) make it quite hard to distinguish the two, as there could be more contrast.

Response: Point accepted. We have enhanced the color contrast in Figure 4.

The authors argue that liganded ROS1 likely forms higher order oligomers, or arrays, which has been suggested in the activation of other RTKs and is a current hot topic of interest in the field (making this paper highly relevant), but they do not model this in their working model (Figure 6). I think this is a big oversight, as someone reading this figure would just think this represents a model for simple ligand-induced dimerization. Figure 6 would also be stronger as a standalone figure if the complex components were labeled or if there was some sort of legend.

Response: We thank the reviewer for this excellent suggestion. In response, we have modified Figure 6 to illustrate how ROS1 is activated through higher-order oligomerization. We have added the complex components and figure legend in our new version.

Lastly, I wonder if the authors tried modeling the dimer using the additional LamG density as a start and then tried to create templates from that to see if it was found in the dataset? The additional density does seem to suggest dimers were present in the sample, but at a lower sampling rate. Given that the monomeric complex structure seems pretty rigid, this may be worth trying if they haven’t already. It may provide a bit more information about the receptor:ligand/co-ligand ‘dimer’.

Response: We thank the reviewer for this excellent suggestion. Following the reviewer's comments, we applied multiple strategies for particle picking, including both template-based and machine learning-based methods, along with extensive 2D and 3D classifications. Despite these efforts, we were unable to determine the structure of the higher-order cROS1/cNEL/hNICOL complex, further indicating that this complex is highly dynamic.

Additionally, the 1:2:1 cROS1/cNEL/hNICOL complex adopts a remarkably elongated conformation, with one dimension of nearly 420 Å, much larger than an 80S ribosome, while the other dimension is only 120 Å. This extreme elongated shape poses significant challenges for particle picking and classification. Capturing the full active state of the ROS1/NEL/NICOL complex will likely require biochemical strategies to stabilize the higher-order complex as well as improvements in cryo-EM image processing methods.

Reviewer #3 (Remarks to the Author):

Overall Assessment

This manuscript reports the cryo-EM structure of the NEL–NICOL–ROS1 complex, providing high-resolution structural insight into ROS1 activation by its recently identified endogenous ligands, NELL2 and NICOL. While ROS1 has long been recognized in oncology as a receptor tyrosine kinase activated by oncogenic fusion proteins, the physiological ligands and activation mechanisms have remained unclear until recent work identified the NELL2/NICOL complex as a key mediator of testis-to-epididymis signaling and male fertility. From a structural biology perspective, no detailed architecture of this ternary complex has been described, making the present work timely and potentially significant.

The study's key strength lies in the novelty of the structural findings, which extend our understanding of ROS1 signaling beyond the oncogenic context and provide a foundation for exploring distinctive modes of RTK activation. The cryo-EM data reveal a ligand–co-ligand–receptor arrangement that offers a structural framework likely to inspire new hypotheses about receptor clustering and signal initiation. The topic will interest structural biologists, reproductive biologists, and the broader RTK signaling community.

Response: We thank the reviewer for the positive assessment of our manuscript.

Several aspects would benefit from further attention to maximize impact and clarity. The methodological description—particularly sample preparation and the selection/heterogeneity of conformational states for cryo-EM analysis—is not yet sufficiently detailed to fully assess reproducibility or the basis of structural interpretation. The linkage to prior biochemical/structural literature could be sharpened; the novelty would be even clearer if the authors positioned their results explicitly relative to earlier models and explained methodological differences that may account for discrepancies. In addition, some mechanistic claims, especially regarding ROS1 activation, would benefit from being phrased with greater caution to strengthen their persuasiveness.

Response: We thank the reviewer for the constructive suggestions. We have expanded the Methods section to provide more details on the sample preparation and cryo-EM analysis and have addressed all other comments in the point-by-point response below.

In summary, this is a potentially high-impact and original study. By adding methodological detail, more explicitly situating the findings within prior literature, and tempering mechanistic proposals where direct evidence is not yet available, the authors can enhance rigor, reproducibility, and interpretability. Addressing the points below should help ensure the work's significance is fully appreciated by a broad readership.

Note on referencing: As the submitted PDF lacks line numbering, specific locations are indicated using figure/panel numbers, section headings, and short quoted phrases rather than page/line numbers.

Major Comments

NICOL-free cNEL

1. According to Extended Data Fig. 1, cROS and cNEL were purified by Ni-affinity chromatography followed by gel filtration on a Superose 6 Increase 10/300 GL column, with peak fractions collected and concentrated. Please indicate whether the gel-filtration step may enrich particular oligomeric states, as this could influence interpretation of subsequent structural analyses.

Response: Good point. We pooled major peak fractions of cROS1 and cNEL after gel filtration, ensuring that most of the protein sample was retained for cryo-EM analysis. We believe that the gel-filtration step does not selectively enrich for particular oligomeric states.

2. For recombinant NICOL-free cNEL, please clarify whether it exists primarily as a monomer or dimer. Under non-reducing SDS-PAGE it appears not to be monomeric, and Extended Data Fig. 1 suggests a major disulfide-linked dimer plus some monomer. Stating this explicitly in the Results would aid interpretation of complex-formation data.

Response: Based on our SDS-PAGE results, recombinant NICOL-free cNEL exists primarily as a disulfide-linked dimer, with a small fraction present as a trimer. This is consistent with previous findings showing that NELL1 exists as both a dimer and a trimer (PMID: 24563467, 21814724). We have clarified this point in the revised manuscript.

3. While the data indicate that NICOL-free cNEL (~240 kDa) is not a monomer, SDS-PAGE alone may not be sufficient to conclude it is a dimer. If feasible, please consider confirming the oligomeric state by mass spectrometry or an equivalent technique to strengthen the conclusion.

Response: We thank the reviewer for the helpful suggestion. We performed SEC-MALS experiments to determine the molecular weight of NICOL-free cNEL under non-reducing conditions. However, due to the poor biochemical behavior of cNEL, SEC-MALS cannot unequivocally determine the molecule weight of cNEL. Since cNEL forms disulfide-linked dimer or trimer that is resistant to SDS, we think SDS-PAGE is an ideal method to sufficiently determine the molecule weight of cNEL. Mass spectrometry of the intact protein would not work in this case because the protein is glycosylated and the exact mass is not known.

4. If NICOL-free cNEL does form a dimer, please clarify whether the LamG–VWC4 interaction occurs between protomers. Indicating whether this is clearly observed, unclear, or unresolved would help readers assess the mechanism.

Response: Since the LamG–VWC4 interaction observed in the cNEL/hNICOL complex does not directly involve NICOL, we suspect that this interaction may also occur in the NICOL-free cNEL dimer. However, we did not observe this interaction in the structure of cNEL alone, because cNEL is highly dynamic in the absence of NICOL. We have briefly discussed this point in the revised manuscript.

cNEL–cROS interaction

5. Extended Data Fig. 1 shows that NICOL-free cNEL used for complex formation with ROS1 was disulfide-linked dimeric rather than monomeric. Explicitly stating this in the Results would contextualize complex formation.

Response: Point accepted. We have noticed this in the revised manuscript.

6. For the cROS–cNEL complex, purified proteins were mixed, separated by gel filtration, and peak fractions were analyzed (Extended Data Fig. 1). Were the SDS-PAGE analyses performed under reducing conditions? If so, what is the band just below cROS—could it be dimeric cNEL or a partially reduced form? Clarifying the analysis conditions and band assignments will aid interpretation.

Response: Yes, we performed SDS-PAGE analysis under reducing conditions. The samples were treated with 50 mM DTT and boiled for 8 minutes before loading onto the gel. We have clarified these analysis conditions in the figure legend. Because the extracellular domain of cROS1 is heavily glycosylated, we believe the lower band likely corresponds to a cROS1 species with incomplete glycosylation. This has been indicated in the revised figure.

7. In the cryo-EM analysis, only the cNEL molecules adjacent to cROS are visible. It remains possible that a cNEL dimer could bind two cROS molecules, particularly if cNEL is more flexible in the absence of NICOL. A brief discussion of this alternative would help frame the observations.

Response: Great point. We have briefly discussed this possible binding mode in the revised manuscript, which could partially explain why cNEL is able to weakly activate ROS1.

8. The gel-filtration profile for cROS–cNEL shows an additional early-eluting peak around column volume ~8. Could this correspond to a cNEL dimer bound to two cROS molecules? Providing an interpretation would be informative for readers considering alternative stoichiometries.

Response: Our new SEC-MALS data showed that the cNEL sample alone exhibits poor biochemical behavior and is prone to aggregation, especially at high concentrations. We therefore believe that the earlier fraction of the cROS1–cNEL sample represents its aggregated form.

cNEL–hNICOL interaction

9. Please discuss, even briefly, structural differences between previously reported NELL2/NICOL complexes and the complex proposed here. Indicating whether differences are methodological, biologically relevant, or both would help contextualize the novelty.

Response: To the best of our knowledge, no structural characterization of the NELL2/NICOL complex has been published to date. Only a portion of the NELL2 structure has been determined in the ROBO3/NELL2 complex.

10. Were there prior predictions or preliminary data suggesting that NELL2 and NICOL might form a covalent complex? If so, summarizing them would provide helpful context.

Response: Good point. Our SDS-PAGE results showed that the cNEL/hNICOL complex is resistant to SDS treatment but can be disrupted by DTT, supporting the model that the complex is covalently linked. We have briefly noted this point in the revised manuscript. To the best of our knowledge, no previous report has indicated covalent complex formation between NELL2 and NICOL in the literature.

11. For the statement, “Consistent with previous findings, our cell-based assays demonstrated that the cNEL/hNICOL complex activates ROS1 much more robustly than cNEL alone,” please verify that the supporting citations are accurate and that the prior findings are represented precisely.

Response: Thank reviewer for pointing out this issue. We have checked this statement and cited the previous paper (PMID: 37095084).

cNEL–hNICOL–cROS interaction

12. Prior studies reported NELL2 as a trimer, with no evidence of a covalent NELL2–NICOL complex. The finding here of a covalent 2:1 NELL2–NICOL complex is highly novel. Explicitly citing the prior work and highlighting the differences would underscore originality.

Response: Good point. NICOL was recently identified as a binding partner of NELL2. We have cited the previous study and briefly discussed the differences in assembly between cNEL alone and the cNEL/hNICOL complex in the revised manuscript.

13. The observation that the NELL2–NICOL complex has two ROS1-binding sites, yet no complex with both sites occupied was observed, may relate to sample preparation—specifically, collecting fractions containing only singly bound complexes (Extended Data Fig. 1). Acknowledging this possibility would temper conclusions about stoichiometry.

Response: We thank the reviewer for raising this critical point. As discussed in the manuscript, the binding of two ROS1 molecules to a single 2:1 cNEL/hNICOL complex would result in severe steric clashes. Therefore, for one 2:1 cNEL/hNICOL complex to simultaneously bind two ROS1 molecules, either ROS1 or cNEL/hNICOL must undergo substantial conformational changes. This mechanism is unlikely but cannot be ruled out, and we have discussed this possibility in the revised manuscript. We also note that further structural studies will be required to capture the fully active state of ROS1.

14. The manuscript states that cNEL is “flexible and monomeric” in the cROS–cNEL complex but “rigid and dimeric” in the presence of hNICOL. Given that the cNEL used for cROS–cNEL complex formation was already dimeric, please reconcile these statements to ensure internal consistency.

Response: Good point! We have modified our statement in the revised manuscript.

15. The steric-clash model preventing simultaneous binding of two ROS1 molecules to a cNEL dimer is interesting. Please specify the extent/nature of the predicted overlap to help readers assess the basis of this conclusion.

Response: Point accepted. We have provided a more detailed description of how the two ROS1 molecules would clash with each other in the revised manuscript. They clash on the head region of ROS1.

ROS1 activation mechanism

16. The proposed activation model involving higher-order multimerization via VWC4–LamG interactions is intriguing but remains speculative based on the current data. Consider moving this to the Discussion, with appropriate caveats, and note explicitly that direct structural evidence is not yet available.

Response: We agree with the reviewer that it is highly speculative to propose that the higher-order multimerization is mediated through VWC4–LamG interactions. Accordingly, we have substantially toned down this model in the revised manuscript. Figures 5A and 5B have been moved to the supplementary materials, and we now briefly mention that the additional density may correspond to the LamG domain from a second 1:2:1 ROS1/NEL/NICOL complex. We explicitly note that this assignment remains uncertain.

17. Please clarify how cNEL and cNEL–hNICOL mutants used in BN-PAGE and cell-based assays were prepared. In particular, indicate whether gel filtration was used to isolate fractions of defined oligomeric state, which is essential for evaluating the activation mechanism.

Response: cNEL was first purified by Ni-affinity chromatography, followed by size-exclusion chromatography using a Superose 6 Increase 10/300 GL column. The major peak fractions were pooled and concentrated for blue native PAGE analysis and cell-based assays. The cNEL–hNICOL wild-type and mutant complexes were expressed and purified using the same procedure. Specifically, hNICOL was co-expressed with either wild-type cNEL or various cNEL mutants in HEK 293F cells, then purified by Ni-affinity chromatography and gel filtration on a Superose 6 Increase 10/300 GL column. The peak fractions were collected and concentrated. Since the major peak fractions represent the majority of the total sample, we do not believe that this purification process selectively enriches a particular oligomeric state for biochemical or functional analyses.

18. The finding that cNEL– Δ LamG–hNICOL fails to form multimers in BN-PAGE and cannot activate

ROS1 despite binding it is mechanistically important. Emphasizing this in the Results and linking it clearly to the proposed model would be helpful.

Response: Thank the reviewer for the great suggestion. We have elaborated this key finding in the revised manuscript.

19. In the section where the authors state that “the cNEL- Δ CC mutant predominantly existed in a monomeric state,” yet BN-PAGE shows multimers: if only monomeric fractions were used for the cell-based assay, this could explain the lack of activation. Clarification would aid interpretation.

Response: Thank the reviewer for making this important point. The reviewer is correct that the cNEL- Δ CC mutant can still form higher-order oligomer, but, as compared to wild-type protein, the ability of the cNEL- Δ CC mutant in higher-order oligomerization becomes much weaker. This explains why cNEL- Δ CC mutant exhibits much weaker activity on ROS1 activation. We suspect that, in addition to the coiled-coil interaction, cNEL can oligomerize through other types of interaction, for example that between VWC4 and LamG domains. We have clarified this issue in the revised manuscript.

20. Since NICOL-free cNEL can form dimers, could it also form higher-order assemblies via VWC4-LamG interactions to activate ROS1? If so, what is the role of hNICOL in this model, particularly with respect to VWC5-NICOL/CC interactions? If unresolved, stating this explicitly would be useful.

Response: We thank the reviewer for raising this important issue. As cNEL alone and the cNEL/hNICOL complex bind ROS1 with similar affinity, and both are capable of oligomerization, it remains unclear why cNEL/hNICOL activates ROS1 much more strongly than cNEL alone. One possible explanation is that the higher-order ROS1/NEL/NICOL complex adopts a conformation that is more favorable for ROS1 activation. In addition, our biochemical and cryo-EM data indicate that NICOL binding enhances the structural stability and biochemical behavior of cNEL, likely increasing the solubility of cNEL. This improved stability and solubility may further contribute to the enhanced biological activity of cNEL. We have briefly discussed this unresolved issue in the revised manuscript and noted that further studies will be necessary to fully elucidate the activation mechanism of ROS1.

21. The mechanism proposes that multiple ROS1 molecules bind a single ligand assembly and approach one another sufficiently for intermolecular phosphorylation. As this arrangement is not directly visualized, acknowledging this limitation would strengthen the Discussion.

Response: Point accepted. In the revised manuscript, we have explicitly noted that the proposed higher-order assembly of the ROS1/NEL/NICOL complex was not directly visualized in our cryo-EM analysis.

22. The manuscript contrasts a “classical RTK” model with a distinct ROS1 activation mechanism. This perspective is potentially correct, but conclusions should be tempered to reflect remaining uncertainties and avoid overstatement.

Response: Point accepted. In the revised manuscript, we have acknowledged that the complete activation mechanism of ROS1 remains unresolved and will require further investigation.

23. The statement that “ROS1 is, to date, the only known RTK that depends on a co-ligand, NICOL, for activation” is not precise, as FGFR activation by FGF requires heparin/heparan sulfate, which function as co-ligands. Please revise accordingly.

Response: Good point. We have removed the “only known” here.

24. The claim that NICOL “stabilizes cNEL into a rigid conformation optimal for ROS1 activation” could be reconsidered. The data may instead support higher-order oligomerization via LamG–VWC4 interactions as the critical determinant. Clarifying this distinction would avoid overinterpretation.

Response: Point accepted. As mentioned above, we have noted in the revised manuscript that the precise role of NICOL binding to NEL in ROS1 activation remains unclear.

Minor Comments

25. For cROS1-H3C-Tsi-His8, hNICOL-H3C-Tsi-His8, cNEL-H3C-Tsi-His8, cNEL-ΔLamG, and cNEL-ΔCC, please provide the amino-acid sequences of linkers/tags appended to the C-terminus in Methods to facilitate reproducibility.

Response: Added.

26. In the section discussing the Drosophila ROS1 homolog Sevenless (Sev) and its ligand Boss, the content is interesting but not directly tied to the manuscript’s core novelty. In addition, the statement “we did not observe cROS1 dimerization under acidic conditions” lacks supporting data. Consider shortening this section or moving it to Supplementary Information.

Response: Good point. We have removed such statements in the revised manuscript.

27. In Fig. 2a and Fig. 3e, the numbering/labeling of VWC domains appears incorrect. Please correct for accuracy and clarity.

Response: Thanks for pointing this out. We have corrected the labeling.

Reviewer #1 (Remarks to the Author):

Review of: “Structural Basis for the Activation of the Chicken ROS1 Receptor by the NEL/NICOL Ligand Complex” by An et al.

The revised draft adds data and at the same time has toned down some speculation that was without supporting data. Although, I would suggest additional toning down on speculation, and reexamining new data. Below are suggestions that could further clarify points for the readers and make for a more polished manuscript. In particular, Fig 6 could be made more impactful to show the thoughts of the authors in their new Discussion section, where they speculate on the impact of NICOL on activation.

We thank the reviewer for the positive assessment of our revised manuscript and the suggestions for further improving our manuscript.

Important control (critical)

Fig 2E, 4c and cell activation throughout (Supp 7, Supp 8, Supp 9, Supp 13). There are no controls for ligand alone (cNEL or cNEL/NICOL) tested to parental cells (HEK or H1299) without cROS1 expression. Unless I missed this? Since NEL is known to interact with another receptor ROBO3 (and potentially additional receptors), downstream ERK could come from many sources. This is a necessary control.

Response: Thank you for pointing out this issue. To address this concern, we performed cell-based activation assays in HEK293T cells with or without cROS1 expression. Upon stimulation with cNEL or cNEL/hNICOL, no ERK phosphorylation was detected in HEK293T cells lacking cROS1, whereas robust ERK activation was observed in cells expressing cROS1 (new Supplementary Fig. 7b). These results rule out the involvement of receptors other than ROS1 in mediating cNEL- or cNEL/hNICOL-induced ERK activation.

Logic missing regarding the comparisons made between cNEL vs cNEL/NICOL induced activation/mechanism:

What is the proposed mechanism for how NICOL changes signaling of cNEL? Reader is left confused about how NICOL affects function. It is ok to state that it remains yet unknown, and that imparted rigidity may play a factor, and show this as hypothesis in Fig 6. Compared to without NICOL.

Response: We agree with the reviewer that the precise mechanism by which NICOL promotes NEL-dependent ROS1 activation remains unclear. Our structural and biochemical analyses show that NICOL binding stabilizes NEL and improves its

biochemical properties. We therefore hypothesize that NICOL enhances the biological activity of NEL by stabilizing its structure and improving its solubility. This point has been incorporated into the revised manuscript: “Our biochemical and cryo-EM data indicate that cNEL exhibits markedly improved biochemical stability and reduced aggregation upon NICOL binding, suggesting that NICOL may act as a chaperone to enhance the biological activity of NEL by stabilizing its structure and improving its solubility. In addition to the chaperone effect, NICOL may facilitate NEL-dependent ROS1 activation through enhancing the formation of the ROS1 active oligomer and positioning multiple ROS1 molecules in a more favorable configuration for activation, although the detail of this mechanism remains unclear.” We have also revised Fig. 6 to reflect this model.

From the data: both ligands (NEL dimer and NEL/NICOL heterotrimer) bind equivalently to ROS1, both form higher order oligomers, both have a foundational dimer of cNEL in the complex. Yet robust signaling requires NICOL. The authors essentially present structures in the manuscript of unliganded ROS1 (Fig 1), ligand bound but inhibited (*partly) ROS1 (NEL only) (Fig 2), and ligand bound active ROS1 (NEL+NICOL) (Fig 3). From the visual presentation of figures the logic rather seems: 1) with no ligand ROS1 has a slightly flexible leg region, 2) with cNEL there is more flexibility, whereas 3) with cNEL plus NICOL, ROS1 has a rigidly linked leg region. This, coupled to cNEL’s oligomerization, seems most supported by the structural analysis as a link to function. Authors should mention some of this directly, as the reader will be drawn to these clear structural differences.

Response: We thank the reviewer for raising this important point. From a structural perspective, we cannot readily explain how binding of NEL or the NEL/NICOL complex alters the flexibility between the “head” and “leg” regions of ROS1, as the ligand-binding site is located far away from the hinge connecting these two regions, and the conformations of the ROS1 “head” are essentially identical in all three states. We therefore speculate that the observed differences in flexibility of ROS1 may have arisen from cryo-EM sample quality. For example, the ROS1/NEL sample exhibits relatively poor biochemical behavior due to the aggregation tendency of NEL, which may hinder cryo-EM structural determination and result in poorly resolved ROS1 “leg” region.

Fig 6. Too much attention paid to the speculative (3 different) ways of oligomerization, and this only focuses on with NICOL present. How does NICOL effect compared to what happens without NICOL? This figure should reflect the new Discussion. Fig6 should show what is supported by data: how NEL interacts with ROS1, how this is NEL dimer (with or without NICOL) bound to 1 ROS1, and that some yet to be determined mechanism (not these 3 speculative but similar ways) leads to higher order clustering. With a “?” showing this resolution

still not there. Real question throughout manuscript is how NICOL affects activity. This should be compared/contrasted to the minus NICOL cluster model.

Response: We thank the reviewer for the excellent suggestions. In response, we have extensively revised Figure 6 in accordance with the reviewer's comments in the revised manuscript.

NEL dimer vs trimer

Authors state that cNEL is a dimer, with or without NICOL present. This is different than mammalian NELL2 or (NELL1) where there is clear evidence that shows NELLs are trimeric: (Nature Communications 2020, PMID: 32198364). Thus, a dimer is unexpected and not consistent with prior reports. This is interesting, as avian NEL is likely different than mammalian NELLs. Authors should point out this interesting finding.

Response: We thank the reviewer for raising this important point. Our purified chicken NEL predominantly forms dimers, but some trimers can be also observed. We notice that in the previous study (Nature Communications 2020, PMID: 32198364), NELL2 was expressed and purified using insect cells, whereas we used a mammalian expression system. Thus, the difference in oligomeric state may arise from either the expression system or species-specific variation, as suggested by the reviewer. This point has been briefly mentioned, and this publication been cited in the revised manuscript: "Interestingly, a previous study indicates that recombinant human NELL1 and NELL2 purified from insect cells predominantly exist as trimers. The observed differences in oligomeric state may therefore arise from differences in the expression system or from species-specific variation." We have also revised Fig. 6 to indicate the co-existence of dimer and trimer of cNEL alone.

Issues with the SEC-MALLs in Fig 5.

Some of the masses predicted exceed a megadalton which is not consistent with the elution volume. These peaks are clearly not at the void volume of column (~15min), yet predicted masses for oligomers far exceed size limitation separated by column. Please explain.

Response: Thank you for raising this concern. Although the stated "optimum" separation mass range for the column is 10,000–600,000, this does not mean that all masses > 600,000 will be in the void volume. There will be a range between the void volume and the elution volume of the optimal maximum (globular) mass that can be observed but not resolved well.

Also, why did authors only show UV for cNEL/hNICOL? The UV signal shows almost none of the protein to be oligomeric. What about UV traces for others? cNEL alone, seems more significant fraction is oligomer (by LS). For cROS1 what is the shown trace? Is it LS or UV, because LS labeled purple, but curve is yellow.

Response: Good point. We initially included only UV trace for cNEL/hNICOL to emphasize that those species were not present at a high population. We have added the UV traces for other samples in the revised manuscript (new Fig. 5).

During cryo-EM grid screening, we observed that cNEL was prone to aggregation at higher concentrations in the absence of NICOL. Consistent with this observation, SEC showed asymmetric peak tailing, and MALS analysis indicated that a substantial fraction of cNEL existed in higher-order oligomeric states.

For cROS1, the trace shown corresponds to the LS. The LS was mistakenly labeled in purple in the previous version. The labeling error has been corrected in the revised manuscript (new Fig. 5). We apologize for this oversight and thank the reviewer for pointing it out.

When one focuses on the linear (non-oligomer) mass predicted of each: it seems cNEL \sim 300 kDa, cNEL/hNICOL \sim 300 kDa, and cROS1 \sim 300 kDa. Is this consistent with authors prediction of dimer cNEL, dimer cNEL+ monomer NICOL, and monomer cROS1? If one then compares these masses to the SEC-MALS of complexes (cROS1/cNEL with mass \sim 900kDa, and cROS1/cNEL/hNICOL with mass \sim 600kDa), this indicates that without NICOL, 2 ROS1' s can actually bind to one cNEL (300+300+300=900). Whereas, with NICOL, only 1 ROS1 can bind (300+300=600). The authors should examine this data for their interpretation. This could fit their structural findings that seem to show increased rigidity with NICOL, and how this might limit a second ROS1 binding event.

Response: We thank the reviewer for raising this critical point. The theoretical molecular weights of cROS1, cNEL, and hNICOL are 209.2 kDa, 95.6 kDa, and 6.7 kDa, respectively. However, both cNEL and cROS1 are heavily glycosylated. For instance, cROS1 contains approximately 30 glycosylation sites. As a result, the actual molecular weights of cROS1, the cNEL dimer, and the 2:1 cNEL/hNICOL complex may all be close to 300 kDa, which is consistent with the SEC-MALS measurements.

We agree with the reviewer that, due to the flexibility of the cNEL dimer, a single cNEL dimer could potentially bind two cROS1 molecules simultaneously. Binding of hNICOL increases the rigidity of the cNEL dimer, thereby preventing association with a second cROS1. We have added the following sentence to the revised manuscript: "Given that cNEL predominantly forms a flexible disulfide-linked dimer, a single cNEL dimer may be capable of binding two cROS1 molecules simultaneously, although such a dimeric complex is not resolved in our cryo-EM analysis."

Minor:

The title: “Structural basis for the Activation of the Chicken ROS1 Receptor by the NEL/NICOL Ligand Complex” implies that the structural basis for activation has been elucidated. However, the certainty described here is rather for how cNEL engages cROS1. Not for activation per say. The exact structural mode of driving activation through clustering remains to be defined.

Response: Good point. We have modified the title as “Structural insights into the Activation of the Chicken ROS1 Receptor by the NEL/NICOL Ligand Complex” in the revised manuscripts.

Should Supplemental Figs be called Extended Data?

Response: Thank you for your concern. We note that many papers published in *Nature Communications* label supplementary figures as “Supplementary Figures” rather than “Extended Data,” and we have followed this convention in our manuscript.

The statements that ROS1 adopts an “extended conformation” is a bit misleading. The N-terminus is in the bent-over conformation as described for Sevenless. “Extended” seems to imply an opening up of this “bent-over” state. The C-terminus ‘leg’ is an extension from this. But one that is not rigid in the absence of ligand. So better described as similar to Sev, N-terminus is folded back, and C-terminal half flexible and dynamic.

Response: We thank the reviewer for this helpful comment. We agree that the term “extended conformation” could be misleading. We have therefore changed the “extended conformation” to “arc-shaped conformation” in the revised manuscript.

Throughout, the N-term domain referred to as HD for helical domain, but this region has been previously referred to as a CATCH domain for (cysteine-rich ALKAL-type coupled Helices) for the drosophila Sevenless receptor.

Response: Point accepted, we have changed HD domain to CATCH domain in the revised version.

Line 23: “ Structural analyses reveal that the 2:1 NEL/NICOL complex further oligomerizes through LamG - VWC4 domain interactions, facilitating the clustering of multiple ROS1 for its activation.”

Does the authors’ “structural analysis” actually “reveal” this? What structural work unambiguously revealed this interaction? Or rather is biochemical

experiments suggestive of this?

Response: Thank you for the comment. Our cryo-EM density suggests a LamG-VWC2 interaction between two cNEL/hNICOL heterotrimers (Supplementary Fig. 12), which is consistent with biochemical experiments showing that cNEL/hNICOL forms higher-order oligomers and that the LamG domain is essential for oligomer formation (Fig. 5b). However, our cryo-EM reconstruction cannot unambiguously resolve this interaction. Accordingly, we have modified the sentence in the manuscript to:
“Structural analyses and biochemical results suggest that the 2:1 NEL/NICOL complexes further oligomerize through LamG-VWC4 domain interactions, facilitating the clustering of multiple ROS1 for its activation.”

Line 77: Despite the present (*presence) of two cNEL protomers in this complex,

Response: Corrected.

Line 106: “A similar D-shaped fold has been observed in the structure of Sevenless (Sev), the Drosophila ortholog of ROS1, indicating the structural conservation of ROS1 across species.” No citation, or structural comparison made here?

Response: Point accepted. The two papers describing Sevenless structures have now been cited here.

Line 246: “SEC-MALS and blue native PAGE (BN-PAGE) results showed ... cNEL/hNICOL and cROS1/cNEL/hNICOL complex can further assemble into higher-order oligomers (Fig. 5a, b).”

But data also shows cNEL and cNEL-ROS1 (without NICOL) can form same higher order oligomers. Therefore, stating this is in isolation is misleading one to believe that cNEL requires NICOL for additional oligomerization.

Response: Good point. We have revised this sentence: “Indeed, both SEC-MALS and blue native PAGE (BN-PAGE) results showed that cROS1 alone predominantly exists as a monomer, whereas the cNEL, cNEL/hNICOL, cROS1/cNEL and cROS1/cNEL/hNICOL complex can further assemble into higher-order oligomers (Fig. 5a, b).”.

Reviewer #2 (Remarks to the Author):

As noted previously, The authors present the first full-length extracellular domain structure of a higher vertebrate receptor tyrosine kinase (RTK) ROS1 in its apo form and bound to the ligand NEL, and a ligand/co-ligand complex NEL/NICOL. This is an exciting finding that provides new insight into oligomerization via interaction with this co-liganded complex. Activation of

RTKs, which are important regulatory signaling molecules and common therapeutic targets, is a topic of great current interest as we are beginning to understand that RTK signaling often has several layers of complexity, and this paper provides some of the first direct structural evidence for clustering mediated by co-ligands, which I think will be of great interest to your readership and something that is widely discussed and cited going forward. Overall, I think the authors have responded adequately to criticisms and questions offered by myself and other reviewers. They provided model and map data that allowed me to answer several of the questions I posed as well. I think the changes to Figure 6 have enhanced the presentations of their conclusions/hypotheses formed by their work. One last comment to the authors: I appreciate their efforts to visualize a receptor:ligand/co-ligand dimer, and just offer one more suggestion to try running *ab-initio* multiple times where you vary the number of classes (up to 10) to see if they might be able to pull out the dimer just in case this has not been tried. This is not a suggestion within the scope of this publication as they have likely tried this and I think the manuscript already provides significant advances that will be of great interest to numerous readers.

Response: Thank you for the positive assessment of our revised manuscript and for the helpful suggestions. We explored multiple strategies for *ab initio* reconstruction, including the approach suggested by the reviewer (varying the number of classes). Despite these efforts, we were unable to resolve a larger complex, likely due to the intrinsic flexibility of this higher-order assembly. We believe that biochemical stabilization of the higher-order ROS/NEL/NICOL complex, or improved sample preparation (for example, using full-length ROS1 for structural studies), will be essential for capturing its structure and will ultimately provide a more complete understanding of how ROS1 is activated by NEL and NICOL.

Reviewer #1:

The authors addressed all of my concerns

Response: we thank the reviewer for supporting the publication.